# NG2 glial cells integrate synaptic input in global and dendritic calcium signals

**Wenjing Sun[1]\*, Elizabeth A Matthews[1], Vicky Nicolas[1], Susanne Schoch[2], Dirk Dietrich[1]\***

[1]Department of Neurosurgery, University Clinic Bonn, Bonn, Germany; [2]Department of Neuropathology, University Clinic Bonn, Bonn, Germany

**Abstract** Synaptic signaling to NG2-expressing oligodendrocyte precursor cells (NG2 cells) could be key to rendering myelination of axons dependent on neuronal activity, but it has remained unclear whether NG2 glial cells integrate and respond to synaptic input. Here we show that NG2 cells perform linear integration of glutamatergic synaptic inputs and respond with increasing dendritic calcium elevations. Synaptic activity induces rapid $Ca^{2+}$ signals mediated by low-voltage activated $Ca^{2+}$ channels under strict inhibitory control of voltage-gated A-type $K^+$ channels. $Ca^{2+}$ signals can be global and originate throughout the cell. However, voltage-gated channels are also found in thin dendrites which act as compartmentalized processing units and generate local calcium transients. Taken together, the activity-dependent control of $Ca^{2+}$ signals by A-type channels and the global versus local signaling domains make intracellular $Ca^{2+}$ in NG2 cells a prime signaling molecule to transform neurotransmitter release into activity-dependent myelination.

## Introduction

NG2-expressing glial cells (NG2 cells) are a large population of self-renewing cells present in substantial numbers in all white and grey matter regions of the CNS throughout development and adulthood (*Nishiyama et al., 1999*; *Butt et al., 2005*; *Trotter et al., 2010*). NG2 cells are precursor cells to all oligodendrocytes, which exclusively provide isolating myelin to axons and are thereby essential for rapid conduction velocity while maintaining small axon diameter in compact brains. Recent evidence suggests that myelination is plastic and activity-dependent and that synaptic transmission to NG2 cells plays a key role in detecting patterns of activity. However, the mechanisms by which neuronal activity is integrated by NG2 cells and transformed into precisely tuned axonal conduction velocities via myelination is not known.

NG2 cells stand out from other glial cells as they receive direct vesicular synaptic input from neighboring axons (*Bergles et al., 2000*; *Lin and Bergles, 2004*; *Jabs et al., 2005*; *Ge et al., 2006*; *Kukley et al., 2007*; *Ziskin et al., 2007*; *Karadottir et al., 2008*; *Kukley et al., 2008*; *Velez-Fort et al., 2010*), and based on these findings, it has recently been proposed that release of neurotransmitter onto NG2 cells may be the key signal mediating activity-dependent myelination and instructing the cells about the need to generate more oligodendrocytes and myelin (*Wake et al., 2011*; *Hines et al., 2015*; *Mensch et al., 2015*; *Koudelka et al., 2016*). NG2 cells are also unique amongst glial cells as they express a large number of voltage-gated ion channels (VGCs) (*Chittajallu et al., 2004*; *Jabs et al., 2005*; *Karadottir et al., 2008*; *De Biase et al., 2010*; *Kukley et al., 2010*; *Clarke et al., 2012*). However, while it is textbook knowledge that neurons use VGCs to integrate synaptic input and to transform it into a pattern of action potentials and associated calcium signals, the functional significance of VGCs and synaptic input in NG2 cells has so far remained obscure. NG2 cells are not capable of firing action potentials (*Bergles et al., 2000*; *Lin et al., 2005*; *Tong et al., 2009*; *De Biase et al., 2010*) and their postsynaptic potentials (PSPs)

\*For correspondence: wsun@uni-bonn.de (WS); dirk.dietrich@uni-bonn.de (DD)

**Competing interests:** The authors declare that no competing interests exist.

are so small (*Jabs et al., 2005*; *Haberlandt et al., 2011*) that it has been questioned whether synaptic input can even achieve a relevant depolarization to recruit VGCs. Furthermore, previous attempts to elicit calcium signals in NG2 cells by synaptic stimulation failed and led the authors to conclude that the synaptic input was too weak (*Velez-Fort et al., 2010*; *Haberlandt et al., 2011*). Thus, at present, we do not know how NG2 cells could transform synaptic input into activity-dependent myelination.

On the other hand, the density of synapses might be massively underestimated, as the low frequency of spontaneous synaptic events may be due to the smaller surface area of NG2 cells when compared to that of neurons (*Kukley et al., 2007*; *Kukley et al., 2008*). Furthermore, the high-input impedance of the delicately thin dendritic branches could result in strong local depolarization even if the response arriving at the soma is small (*Sun and Dietrich, 2013*).

Deciphering how NG2 cells can integrate and process synaptic input is of fundamental importance for understanding brain development and for improving re-myelination of damaged white matter. Therefore, in this study, we investigated whether synaptic depolarizations recruit VGCs in the somatic or dendritic compartment of NG2 cells. We report that NG2 cells possess a dedicated signal integration machinery, different from that of neurons and other types of glial cells, and generate local dendritic or global $Ca^{2+}$ responses depending on the pattern and type of incoming activity.

## Results

We first set out to obtain a measure of the maximal level of synaptic depolarization of NG2 cells. To this end, we recorded DsRed-expressing NG2 cells in CA1 stratum radiatum from an NG2-DsRed mouse line in current-clamp mode (*Figure 1A*) and optimized the electrical stimulation of Schaffer collaterals to estimate the maximally attainable postsynaptic depolarization. In four NG2 cells resting at −85 mV, we recorded a substantial peak depolarization of PSPs up to −35.4 ± 8.6 mV (n=4, *Figure 1B*).

To further explore whether PSPs recruit VGCs and to obtain precise control of the degree and timing of intracellular depolarization, we induced mock PSPs by current injection. Realistic mock PSPs were generated by deriving the current injection waveform from miniature excitatory postsynaptic currents (EPSCs) recorded in NG2 cells representing the average conductance change in response to release of a single glutamate-filled vesicle. Upon increasing the amplitude of the injected current waveform (*Figure 1C*, grey shaded area) the response increased concomitantly, and we observed an obvious shortening of the voltage response (black line) reminiscent of the activation of VGCs (*Figure 1C*). To analyze the impact of VGCs, we compared these mock PSP responses to tiny hyperpolarizations, which we produced by injecting very small negative mock currents designed to avoid activation of VGCs. We arithmetically scaled up such small voltage responses by the ratio of the injected current strengths (see Materials and methods for details) to create a hypothetical, large response unaffected by VGCs (dashed grey line, *Figure 1C*).

To dissect the contribution of different types of VGCs to the shortening of mock PSPs, we bath-applied established blockers of VGCs (*Figure 2A,B*). The A-type K⁺-channel blocker 4-aminopyridine (4-AP) potently reversed the shortening of the mock PSP and significantly increased the half-width from 2.4 ± 0.2 ms to 3.8 ± 0.5 ms (n=4, Paired *t*-test). In addition, 4-AP increased the amplitude of the mock PSP from 65.8 ± 2.1 mV to 78.7 ± 4.3 mV (significantly different, s.d., Paired *t*-test). By contrast, blocking Na⁺ channels by using tetrodotoxin (TTX) significantly reduced the amplitude from 66.1 ± 0.9 mV to 57.9 ± 1.1 mV (n=7, Paired *t*-test), and also slightly increased the half width of PSPs (from 2.3 ± 0.3 ms to 3.0 ± 0.2 ms, s.d., Paired *t*-test). The latter is likely secondary to the reduced amplitude, which results in reduced recruitment of A-type channels. Delayed rectifier K⁺ channels are not recruited by individual mock PSPs because tetraethylammonium (TEA) affected neither their amplitude nor their half-width (n = 6, Paired *t*-test).

In order to identify the membrane potential range in which VGCs modulate synaptic potentials, we varied the amplitude of mock current injections in the absence or presence of ion-channel blockers. We quantified VGC-induced changes of mock PSPs by dividing their amplitude and half-width by the amplitude and half-width of scaled passive responses (negative current injection as above) and plotted the resulting half-width- and amplitude-ratio against the membrane potential reached at the peak of the recorded mock PSPs (*Figure 2C–E*). These plots clearly showed that in the absence of channel blockers, the durations of mock PSPs were increasingly shortened when the membrane

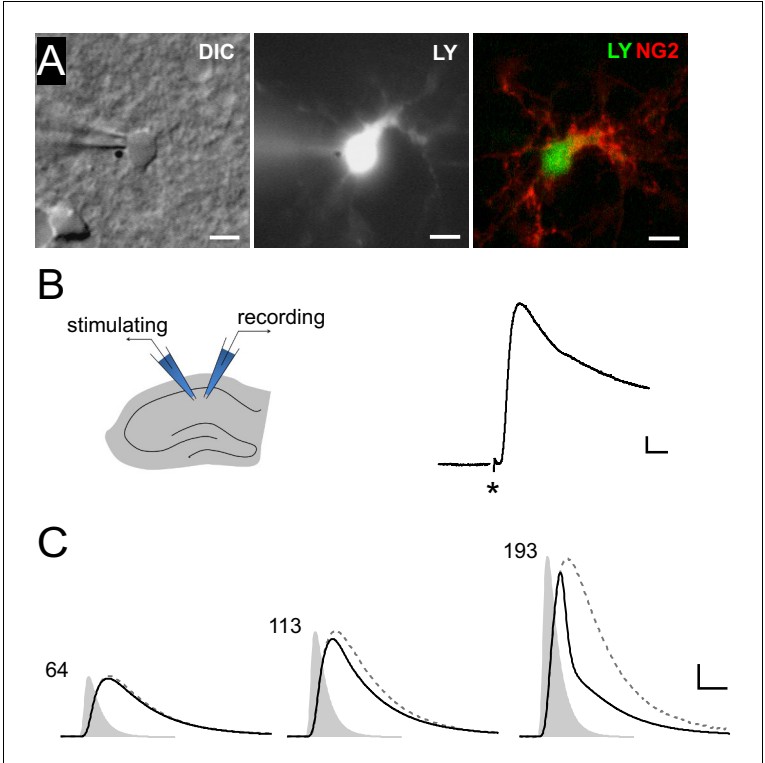

**Figure 1.** Synaptic potentials in NG2 cells can recruit voltage-gated ion channels. (**A**) Identification of NG2 cells. Wide-field images of a lucifer yellow-filled cell expressing DsRed (not shown) in the acute slice preparation (left: DIC, middle: epifluorescence). Post-recording anti-NG2 staining confirmed the cell's identity (right). Scaling: 5 μm. (**B**) Left: Experimental configuration used for Schaffer-collateral stimulation and whole-cell recording of postsynaptic potentials (PSPs) in NG2 cells. Right: Example PSP recording with the asterisk showing the time of the stimulation. Scaling: 5 mV, 5 ms. (**C**) Large mock PSPs (black lines) are shortened when compared to small mock PSPs (dashed lines, scaled for comparison). Injected currents derived from miniature excitatory postsynaptic currents (EPSCs) are shown as grey shaded areas and the numbers indicate their amplitudes in terms of glutamate quanta. The kinetic differences between the waveform of the injected current and the small mock PSP in the left panel (64 quanta) is due to filtering by the membrane capacitance. Scaling: 10 mV, 3 ms.

depolarization exceeded $-60$ mV (*Figure 2D*, black line), consistent with the low-voltage activation range of A-type channels. Near the maximally attainable average depolarization of $-16.5 \pm 1.0$ mV, the half-width ratio of mock PSPs was substantially shortened to $47.1 \pm 3.4\%$ (*Figure 2D*). This shortening is almost exclusively due to the action of A-type K$^+$ channels, whereas blocking Na$^+$ and delayed-rectifier K$^+$ channels did not detectably change the duration of mock PSPs (*Figure 2C,D*). In contrast to the half-width, the amplitude of mock PSPs was only slightly reduced, as indicated by an amplitude ratio near 1 even for large PSPs (*Figure 2E*, black line). This linear scaling of the amplitude of mock PSPs in control conditions is a result of a balanced interplay of fast Na$^+$ and A-type K$^+$ channels (*Figure 2C,E*): if A-type or Na$^+$ channels are blocked, the amplitude of mock PSPs is increased or decreased by ~15% and ~30%, respectively. Taken together, these results demonstrate a prominent role of A-type K$^+$ channels in shaping synaptic input and suggest that even medium-sized synaptic input ($\Delta$ 20–30 mV) is accelerated by A-type K$^+$ channels, while the amplitude of synaptic depolarizations faithfully encodes the input strength due to mutual antagonism of A-type K$^+$ and fast Na$^+$ channels.

In neurons, voltage-gated K$^+$ and Na$^+$ channels are responsible for generating an action potential waveform upon integration of synaptic input. Action potentials are potent openers of voltage-gated calcium channels (VGCCs) and elicit prominent calcium signals. Since NG2 cells do not fire action potentials (*Bergles et al., 2000*; *Lin et al., 2005*; *Tong et al., 2009*; *De Biase et al., 2010*), we wondered whether PSPs may directly trigger intracellular calcium signals, and therefore combined 2-

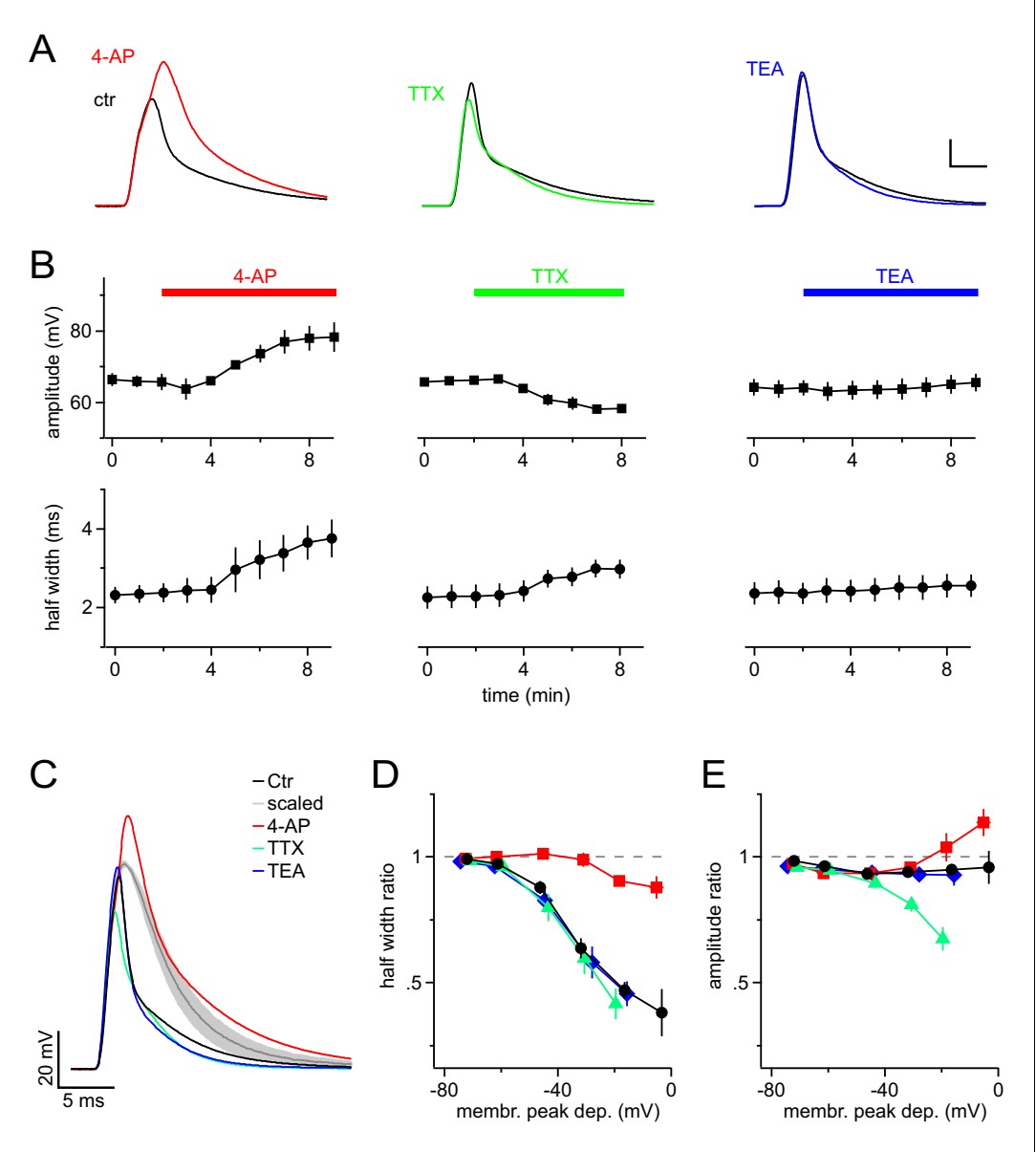

**Figure 2.** A-type K$^+$ channels shorten mock PSPs in NG2 cells. (**A**) Blocking A-type K$^+$ channels with 4 mM 4-AP strongly broadens mock PSPs and increases their amplitude, while blocking Na$^+$ channels with 1 µM TTX slightly reduces their amplitude. Application of 10 mM TEA to inhibit delayed-rectifier K$^+$ channels shows no effect. Scalings: 15 mV, 3 ms. (**B**) Time-course of the average amplitude and half width of the mock PSPs during application of ion channel blockers. (**C**) Superimposed mock PSPs recorded after pre-treating cells with channel blockers. Cells were kept at −85 mV and mock current injection amplitude was chosen based on small passive PSP responses in each cell, such that they would be depolarized to −15 mV if there were no voltage-gated channels present. The grey line represents the average of the scaled passive responses for all examples shown (shaded area indicates standard error of the mean (SEM), n=4 cells). Note the difference between the grey and the other lines, illustrating the reproducibility of the effect of ion channels on the mock PSPs across different NG2 cells. (**D**) A-type channels shorten mock PSPs when NG2 cells are depolarized above −60 mV. The width of mock PSPs is normalized on the width of a passive mock PSP recorded in the same cell and plotted against the membrane potential reached during the peak of the PSPs. Gradually increased current injections were used to achieve larger depolarizations. Note that 4-AP almost completely removes shortening of PSPs. (**E**) A-type K$^+$ channels and fast Na$^+$ channels affect the amplitude of mock PSPs in opposite ways and cancel each other's effect. Plot as in (**D**), except that the normalized amplitude is plotted on the left axis. Note that the control curve (black line) stays close

*Figure 2 continued on next page*

*Figure 2 continued*

to unity showing the fine balance between TTX- and 4-AP-sensitive channels. For D & E, n=14, 5, 7 and 6 cells for control, 4-AP, TTX and TEA group, respectively.

photon $Ca^{2+}$ imaging with our mock PSP stimulation paradigm. During injection of a robust stimulus (10 mock PSPs at 100 Hz), we monitored the fluorescence of the calcium indicator dye Fluo-4 (200 µM) with line-scans across proximal dendrites and the soma of NG2 cells (*Figure 3A–D*). Indeed, we detected clear calcium transients in the soma and proximal dendrites ($3.9 \pm 1.3\%$ and $12.7 \pm 1.4\%$, *Figure 3E*). These $Ca^{2+}$ signals required the activation of VGCCs, as they were highly sensitive to a cocktail of VGCCs blockers (*Figure 3F,G*). Further, the high sensitivity of $Ca^{2+}$ signals to combined application of SNX-482 (1 µM, R-type VGCCs blocker, *Newcomb et al., 1998*) and TTA-P2 (30 µM, T-type VGCCs blocker, *Choe et al., 2011*) suggested that they are mediated by low-voltage-activated $Ca^{2+}$ channels (*Figure 4A and B*). In contrast, $Na^+/Ca^{2+}$ exchangers, which were previously shown to mediate pharmacologically induced $Ca^{2+}$ signals, are not involved in mock EPSP-induced $Ca^{2+}$ signals as blocking these exchangers with KB-R7943 (10 µM, *Iwamoto et al., 1996*) did not decrease their amplitude (*Figure 4A and B*). $Ca^{2+}$ signals decayed back to baseline in an exponential manner ($\tau_{decay}=3.16 \pm 0.5$ s, n=9, see Materials and methods, *Figure 4C*) and could be reproducibly elicited over a period of more than 12 min (*Figure 4D*, n=5).

Because somatic current injection regularly produced a calcium response in proximal dendrites, we analyzed the spatiotemporal distribution of calcium in NG2 cells in more detail by combining 2-photon microscopy and fast frame acquisition (30 Hz). Scanning areas and focus planes were selected to almost fully contain at least one primary dendrite which extended up to ~35 µm from the soma (*Kukley et al., 2010*, *Figure 5A*). Injection of 10 mock PSPs as above generated an almost immediate increase in fluorescence throughout all scanned dendrites, which decayed on the order of several seconds (*Figure 5B and D*). We quantified this signal within regions of interest (ROIs) covering successive 5 µm segments along dendrites (*Figure 5C*). This analysis showed that the $Ca^{2+}$ signal amplitude remained constant throughout the dendritic tree, or even increased towards the periphery (*Figure 5D and E*), and that the signal started almost simultaneously, irrespective of the distance from the soma (*Figure 5F*). In particular, the spread of the $Ca^{2+}$ signal was much faster than expected from pure diffusional propagation from the soma into dendrites (*Figure 5F*). Together with the undiminished amplitude and steep rise of the dendritic $Ca^{2+}$ signal, the data suggest that the membrane depolarization of the mock PSPs rapidly spreads throughout the dendritic tree and opens local dendritic VGCCs. The minimal delay and the rapid rise of dendritic $Ca^{2+}$ signals in NG2 cells is comparable to the fast dendritic $Ca^{2+}$ responses of CA1 pyramidal neurons recorded and stimulated under identical conditions (*Figure 5G and H*), suggesting that NG2 cells are capable of similar dendritic signal processing.

A train of 10 large PSPs may not happen very frequently under physiological conditions because it requires a large number of presynaptic neurons to fire in a highly synchronized manner. We therefore stimulated NG2 cells with a more realistic pattern of synaptic input. We allowed firing of presynaptic neurons to deviate from perfect synchrony and modeled a temporal distribution of presynaptic activity using a Gaussian distribution with a standard deviation of 25 ms (*Figure 6A*). We progressively increased the number of injected Gaussian-distributed quanta (Q) and monitored intracellular calcium levels with 2-photon microscopy in line-scan mode. In the control group, $Ca^{2+}$ responses were seen only rarely (*Figure 6B*). By contrast, when blocking A-type $K^+$ channels, we observed larger depolarizations often carrying prominent spikelets (in five out of seven cells, average amplitude $40.2 \pm 6.7$ mV), which were accompanied by substantial intracellular $Ca^{2+}$ elevations (*Figure 6C*). When plotting the peak depolarization against the current injection strength (being similar across groups, *Figure 7A*), voltage responses in the control and 4-AP groups were indistinguishable as long as they did not exceed ~−45 mV, consistent with the threshold for recruitment of A-type channels as determined above. As expected, larger current injections (injections equivalent to >600 glutamate quanta, see Materials and methods) much more effectively depolarized NG2 cells in the presence of 4-AP (*Figure 7A*). To discriminate $Ca^{2+}$ responses from non-responding trials, we analyzed the line-scan profiles with a template-fitting algorithm. This analysis revealed (*Figure 7A*) that for both groups $Ca^{2+}$ responses (filled symbols) are much more frequent if NG2 cells are

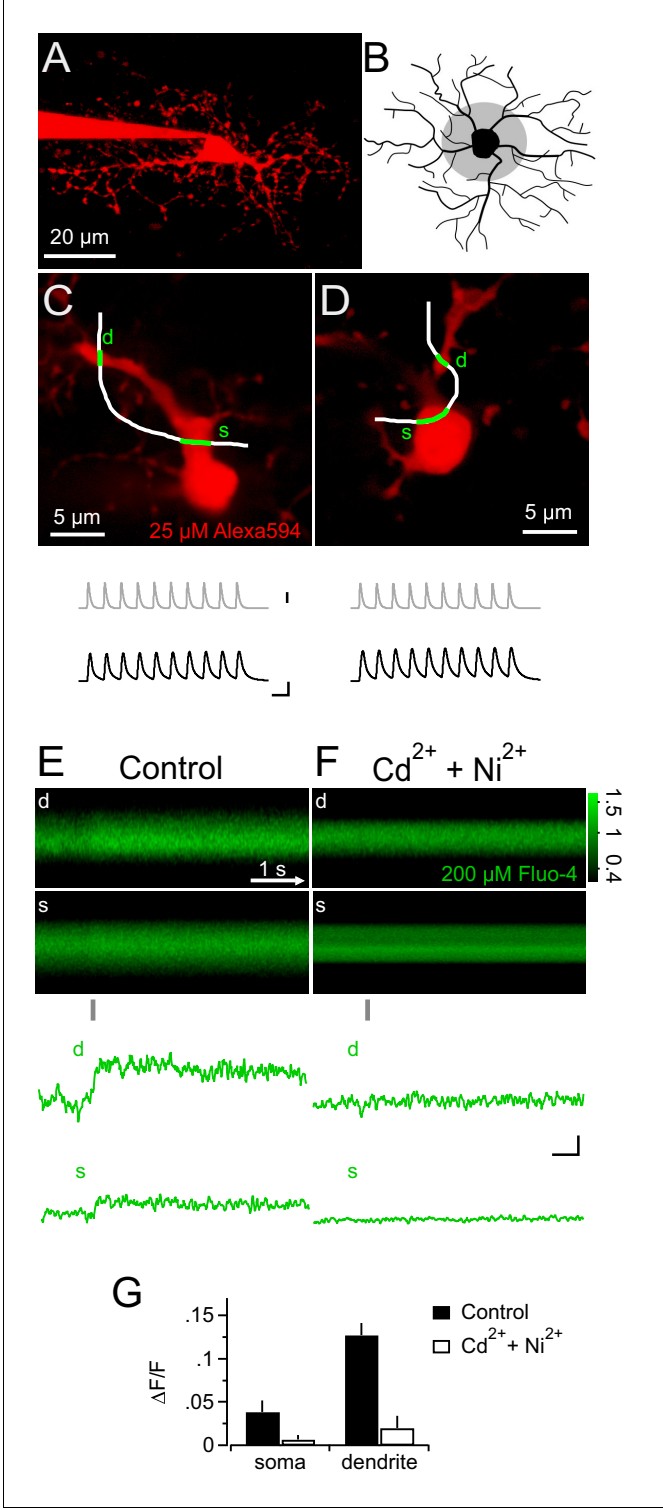

**Figure 3.** Synaptic depolarizations trigger Ca$^{2+}$ signals in NG2 cells by recruiting VGCCs. (**A**) Maximum intensity projection from a 2-photon image stack of an Alexa 594-filled NG2 cell during recording in a hippocampal slice. (**B**) Grey shaded area indicates region used for selecting line-scans. Lines were scanned within the inner one third of the dendritic tree. (**C**) Frame-scan (Alexa 594) illustrates configuration of freehand line-scan (white line) through soma(s) and dendrite (d) for a control group cell (n=7). Grey and black traces show the injected current waveform and the voltage response, respectively. The average amplitude of the first PSP was 78.0 ± 2.9 mV. Scaling: grey trace, 1000 pA; black traces, 40 mV, 10 ms (also apply to **D**). (**D**) As in (**C**) but for an NG2 cell bathed 10–15 min in

*Figure 3 continued on next page*

*Figure 3 continued*

300 µM $Cd^{2+}$ and 200 µM $Ni^{2+}$ (n=5). The average amplitude of the first PSP was 89.5 ± 1.6 mV. (**E**) Rectangular images show the temporal sequence of the somatic and dendritic parts of the freehand line-scan of the cell shown in (**C**). Grey bar below the images indicates the time and duration of current injection. Both images show a clear fluorescence increase following current injection. Traces at the bottom (line-scan profiles) show spatial averages of line-scans of the Fluo-4 channel over time (d, dendrite, s, soma). Note the clear fluorescence increase after the current injection and that the current injection is very brief compared to the calcium signal and happens only during its rising phase. Line-scans and line-scan profiles are the average of three repetitions. (**F**) As is (**E**) but for an NG2 cell bathed 10–15 min in 300 µM $Cd^{2+}$ and 200 µM $Ni^{2+}$ (n=5). Note the absence of an indicator response after current injection in the presence of these VGCCs antagonists. Scalings also apply to E: 0.1 ΔF/F, 0.5s. (**G**) $Ni^{2+}$ and $Cd^{2+}$ potently block the peak amplitudes of both the somatic and the dendritic calcium transient.

depolarized beyond approximately −45 mV, suggesting that blocking A-type channels facilitates $Ca^{2+}$ signals by allowing for stronger depolarizations. $Ca^{2+}$ signals were not only significantly more frequent in the presence of 4-AP (control: 9/40; 4-AP: 22/43; Fisher's exact test, *Figure 7B*), but the responding trials also showed a significantly larger amplitude: 10.1 ± 0.7% and 14.1 ± 1.5% in the control and 4-AP groups, respectively (Student's *t*-test, *Figure 7C*). Further analysis showed that the fraction of trials that responded with a $Ca^{2+}$ signal steadily increased towards unity with the quantal content of the stimulus in both groups. However, 4-AP treatment left-shifted this curve, lowered the 50% threshold from ~1200 to ~600 quanta, and thereby strongly increased the propensity of NG2 glial cells to generate $Ca^{2+}$ signals (*Figure 7D*). Interestingly, re-plotting the fraction of responding trials against the peak depolarization minimizes the difference between the two groups seen when plotting the fraction against injection strength (*Figure 7E*), further indicating that 4-AP does not change the voltage dependence of the $Ca^{2+}$ signal but does change the recruitment of $Ca^{2+}$ channels. The strength of the template-fitting algorithm lies in its ability to identify clear responses unambiguously, but as a result, 'non-responding' trials do not enter the analysis. To characterize the whole population of NG2 cell dendrites, we therefore undertook an additional analysis including all dendrite recordings. We binned trials according to stimulus strength, as illustrated in *Figure 7D*, but calculated the average ΔF/F ratios across all trials. This analysis revealed a small but statistically significant $Ca^{2+}$ signal in the control group and also showed that dendritic $Ca^{2+}$ levels linearly increased with input strength when A-type $K^+$ channels were blocked (*Figure 7F*). In summary, these results indicate that A-type $K^+$ channels tightly gate the generation of $Ca^{2+}$ signals in response to synaptic input to NG2 cells.

Dendrites are likely to receive the majority of synapses as they provide the major part of the cell's surface area (*Kukley et al., 2008*; *Sun and Dietrich, 2013*). Thus the largest fraction of synaptic currents originates in dendrites and its integration is difficult to study with somatic current injection. To explore dendritic integration of synaptic input by NG2 cells, we employed 2-photon-based MNI-glutamate uncaging. This technique allowed us to produce very localized, diffraction-limited, and brief (0.65 ms, at 720 nm) photo-release of glutamate onto dendritic segments, mimicking the rapid liberation of glutamate from presynaptic vesicles. By pointing the scanner at segments in the outer two thirds of the NG2 cell dendritic tree (see scheme in *Figure 8A*, grey shaded area and *Figure 8B*), we produced uncaging EPSPs (uEPSPs, *Figure 8C*), which compared well in amplitude and kinetics to the small spontaneously occurring fast EPSPs in NG2 cells (*Figure 8D,E*). Generating six uEPSPs within the same dendritic subsegment almost simultaneously produced a compound uEPSP with an amplitude of 3.2 ± 0.9 mV, whereas arithmetic summation of the six sequentially acquired responses from the same individual positions amounted to 4.4 ± 1.3 mV (*Figure 8C* bottom panel, **F**, **G**).

We next asked whether dendrites may generate $Ca^{2+}$ signals more readily in response to local synaptic input and, in particular, whether they might respond to the additional activity of a few synapses if many other synapses were already active throughout the whole dendritic tree. To address this question, we employed mock PSPs to simulate a large compound synaptic response, sized to remain below the threshold of $Ca^{2+}$ responses, and then added six active synapses, co-localized within one dendritic subsegment, by local 2-photon glutamate uncaging (>10 µm from soma, *Figure 9A*). Under control conditions, the addition of six uEPSPs to the mock PSP hardly increased the depolarization (−40.8 ± 1.4 mV vs −39.6 ± 1.3 mV, *Figure 9B*), as expected from the small

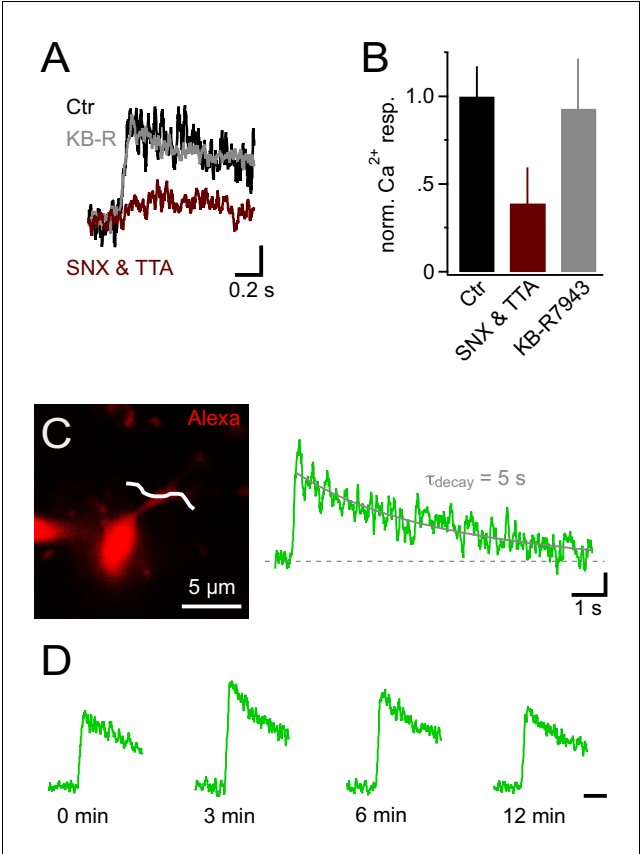

**Figure 4.** Ca$^{2+}$ signals in NG2 cells are mainly generated by low-voltage-activated Ca$^{2+}$ channels. (**A**) Ca$^{2+}$ signals evoked by trains of 10 PSPs (100 Hz) calculated from line-scans across proximal dendrites after 10 min in control or blocker-containing solution. Note the pronounced inhibitory effect of 1 µM SNX-482 (R-type VGCC blocker) and 30 µM TTA-P2 (T-type VGCC blocker) (maroon trace). The amplitudes of Ca$^{2+}$ signals in each experiment have been normalized on the initial Ca$^{2+}$ signals (0 min) obtained in the same cell before application of blockers or control solution (initial response not shown). Blocking Na$^+$/Ca$^{2+}$ exchangers with 10 µM KB-R7943 does not affect Ca$^{2+}$ signals (grey trace). Scaling: 20% of 0 min Ca$^{2+}$ signal peak amplitude, 0.2 s. (**B**) Summary bar graph showing that in SNX-482 and TTA-P2 only 39.1 ± 20.4% of the signal seen in control solution remains after 10 min (n=11, 5 and 5 for control, SNX&TTA and KB-R7943 groups, respectively). (**C**) 2-photon scan of proximal dendrite of NG2 cell used for recording the Ca$^{2+}$ signal shown on the right. The white line indicates the position of line-scan. The long line-scan profile on the right illustrates the decay of the signal. The grey line indicates the exponential fit yielding $\tau_{decay}$. Scaling: 0.05 ΔF/F, 1 s. (**D**) Ca$^{2+}$ signals can be repetitively evoked in NG2 cells. Line-scan profiles show recordings from the same dendritic location. The amplitudes of Ca$^{2+}$ signals at each time point have been normalized on the initial Ca$^{2+}$ signals (0 min). Scaling: 0.5 s.

amplitude of the uEPSP, and also did not generate an increased calcium response in the stimulated dendrite (6.0 ± 1.2% vs 8.8 ± 2.5% ΔF/F, one-way ANOVA for correlated samples, Tukey HSD, *Figure 9B and D*). However, the situation was strikingly different if we inhibited A-type K$^+$ channels by application of 4-AP. The simulated global synaptic input of the same strength applied alone or together with the six uEPSPs produced a comparable depolarization of the soma (−36.5 ± 1.4 mV vs −34.0 ± 1.7 mV, *Figure 9C*). In addition, the mock PSP on its own was not associated with a clear Ca$^{2+}$ response. However, adding local uEPSPs significantly boosted the Ca$^{2+}$ signal in the stimulated dendrite (6.7 ± 1.3% vs 14.3 ± 3.3% ΔF/F, one-way ANOVA for correlated samples, Tukey HSD, *Figure 9C,D*). Taken together, these data show that dendritic intracellular Ca$^{2+}$ levels in NG2 cells indicate the activity of local synaptic input despite the small amplitude of the locally induced synaptic potential.

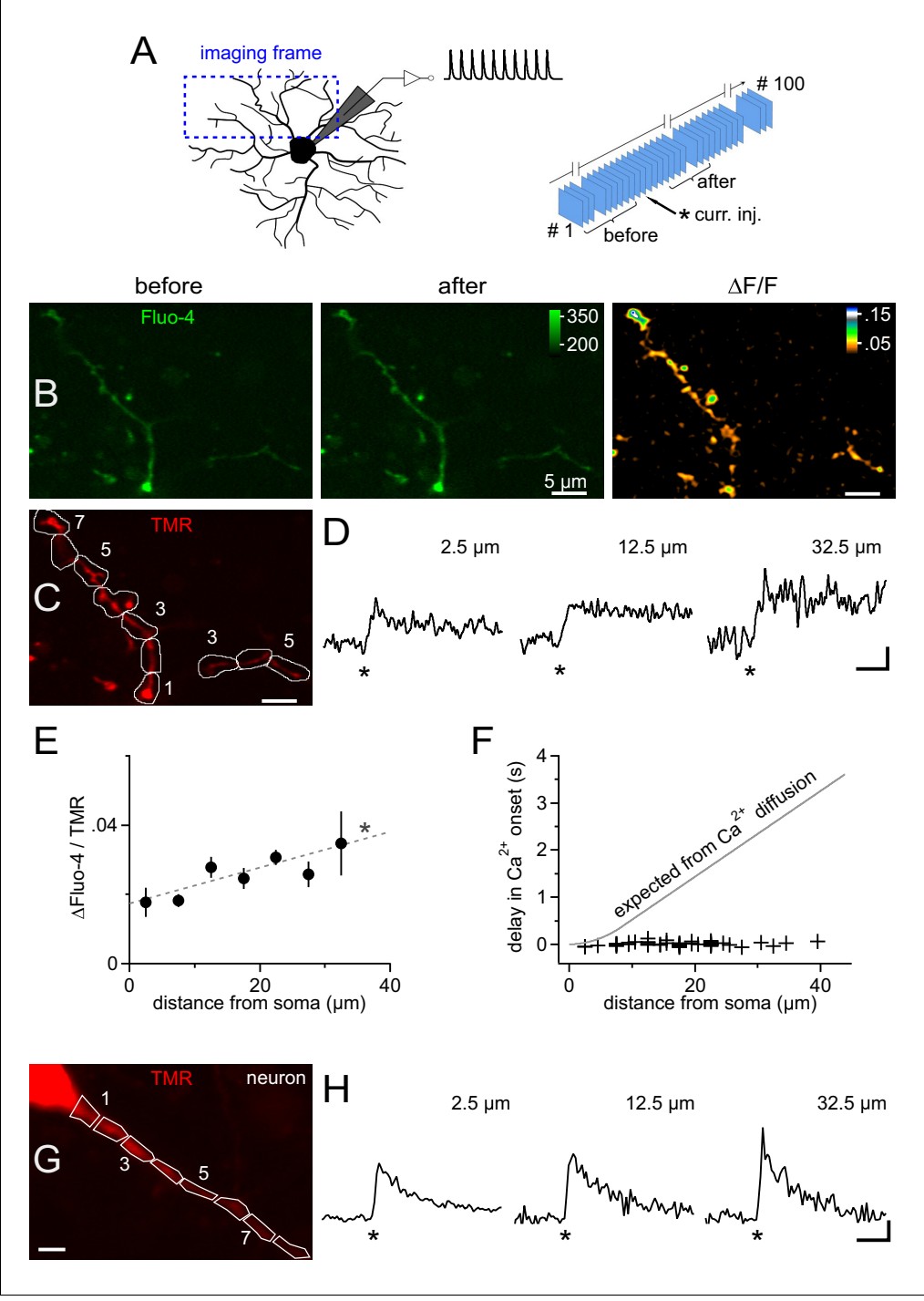

**Figure 5.** Ca$^{2+}$ transients are global and instantly occur in distal dendrites of NG2 cells. (**A**) Left: Scheme showing the strategy for selecting frames for fast time-lapse acquisition of calcium indicator dye-filled NG2 cells stimulated in current clamp mode (inset on top shows waveform of injected current, as used for **Figure 3**). Frame-scans include very proximal and terminal dendrites of NG2 cells. Right: Sequence of image acquisition (100 @ 30 Hz). Curly brackets denote frames being averaged for the analysis shown in panels (**B**–**F**). (**B**) Average frame-scan of Fluo-4 fluorescence before (left) and after (middle) mock PSPs. Note the increase in brightness along the entire dendrite. This is better seen in the right image displaying the normalized difference in fluorescence ($\Delta F/F$). (**C**) Corresponding frame-scans of the tracer tetramethylrhodamine-biocytin (TMR). ROIs used for analysis are drawn consecutively with 5 µm increments along the dendrite. ROIs (numbered sequentially) with similar distances to the soma edge are binned for quantification in (**E**). (**D**) Calcium signals rise rapidly and with undiminished amplitude at

*Figure 5 continued on next page*

*Figure 5 continued*

proximal and distal locations. Traces show the time course of the mean fluorescence of ROIs at three different distances expressed as a volume-normalized signal (Fluo-4/TMR) (n=13, 34 and 5 ROIs, respectively). Asterisks indicate time of current injection. Scaling: 0.02 Fluo-4/TMR, 500 ms. (E) Average $Ca^{2+}$ transient amplitudes slightly increase with the distance from the soma across the dendritic tree (linear correlation test, n=141 ROIs from 12 time-lapse series of seven NG2 cells, $R^2$=0.08; dashed grey line). Peak amplitude values of ΔFluo-4/TMR were binned according to distance from soma (n=13, 22, 34, 28, 25, 13 and 5 ROIs, respectively). (F) Minimal delay of calcium signals even distally from the soma. Onset of $Ca^{2+}$ signals was determined for each individual ROI using a template-fitting algorithm (see Materials and methods). If the calcium signal were to propagate by pure diffusion from the somatic site of current injection, substantial delays for remote positions would be expected as denoted by the grey line (see Materials and methods). In order to obtain a precise estimate of the onset of the signal, a detection threshold was chosen (3.0, see Materials and methods) to select signals very clearly standing out of the recording noise (44 ROIs from nine time-lapse series of seven cells). (G) Primary dendrite of a CA1 pyramidal neuron used for a comparative analysis of the dendritic occurrence of $Ca^{2+}$ signals. ROIs were defined as in (C). Scaling: 5 µm. (H) $Ca^{2+}$ signals generated in dendrites of CA1 pyramidal neurons, which are known to express dendritic VGCCs, display fast rise times and minimal delays similar to those observed for NG2 cells. Note, however, that neuronal $Ca^{2+}$ signals decay more rapidly. Neurons were bathed in TTX to prevent action potentials and to make the experiment more comparable to NG2 cells (showing no action potentials). As for NG2 cells, neurons were stimulated with 1 10 pulse 100 Hz train of mock PSPs (n=3). Asterisks indicate time of current injection. Scaling: 1.0 Δ(Fluo-4/TMR)/(Fluo-4/TMR), 500 ms.

Considering the pivotal and powerful role of the activity of A-type $K^+$ channels in allowing calcium signals to be triggered in NG2 cells, we asked whether synaptic input itself may bring a relevant fraction of A-type channels into inactivation. When plotting the half-width of the mock PSP voltage responses in a train (10 PSPs at 100 Hz), we found a clear increase of 26.3 ± 3.3% (n=7, *Figure 10A, B*). This progressive broadening of PSPs is very likely to reflect an increasing fraction of inactivated A-type $K^+$ channels as this broadening was not observed in the 4-AP-treated group (n=8, *Figure 10A,B*). Such an increase in half-width is very likely to enhance $Ca^{2+}$ entry via an increased tail current during the repolarization phase ($Ca^{2+}$ channels opened, prolonged period with large and increasing driving force for $Ca^{2+}$). Indeed, with broadened PSPs in the 4-AP treated group, we observed a significantly increased $Ca^{2+}$ response (*Figure 10C and D*, 12.7 ± 1.4% vs 24.2 ± 4.2% in the control and 4-AP groups, respectively, Student's *t*-test). Thus, the data suggest that use-dependent inactivation of A-type channels renders a high frequency train of synaptic input that is effective in causing $Ca^{2+}$ responses in NG2 glial cells. Amplification of mock PSP-induced $Ca^{2+}$ signals was also seen when voltage-gated $K^+$ channels were blocked by the inclusion of $Cs^+$ in the pipette solution (n = 10, *Figure 10C and D*, 12.7 ± 1.4% vs 30.5 ± 9.3% in control and $Cs^+$ groups, respectively, Wilcoxon-Mann-Whitney test). Furthermore, the pipette solution also contained 1 µM of the SERCA blocker thapsigargin and thus showed that $Ca^{2+}$ stores are not important for mock PSP-induced $Ca^{2+}$ signals in NG2 cells.

While the above experiments demonstrated that synaptic input can lead to large depolarization of the soma of NG2 cells (*Figure 1B*), all $Ca^{2+}$ signals reported so far were evoked by direct current injection. Therefore, we designed an experiment to show that synaptic transmitter release can trigger local $Ca^{2+}$ signals in NG2 cell dendrites. We placed a small glass electrode in the immediate vicinity of a small dendritic branch (*Figure 10E*) and elicited a brief train of EPSPs (five or six stimuli at 100 HZ, *Figure 10E*). As seen with local glutamate uncaging, these dendritically induced EPSPs were recorded with small amplitudes by the electrode attached to the soma (7.1 ± 1.1 mV, n=4). Nevertheless, in the nearby dendrite, these EPSPs were accompanied by a clear $Ca^{2+}$ signal with kinetics and size similar to those shown above for $Ca^{2+}$ signals induced by current injection or glutamate uncaging (21.1 ± 3.4%, n=4, *Figure 10E*, 1 µM thapsigargin in the pipette). Furthermore, as argued above, this observation also suggested that the local depolarization in the dendrite is much larger than that observed at the soma and that there is strong voltage attenuation along a dendrite. More importantly, this experiment substantiates the physiological relevance of $Ca^{2+}$ signals in NG2 cells.

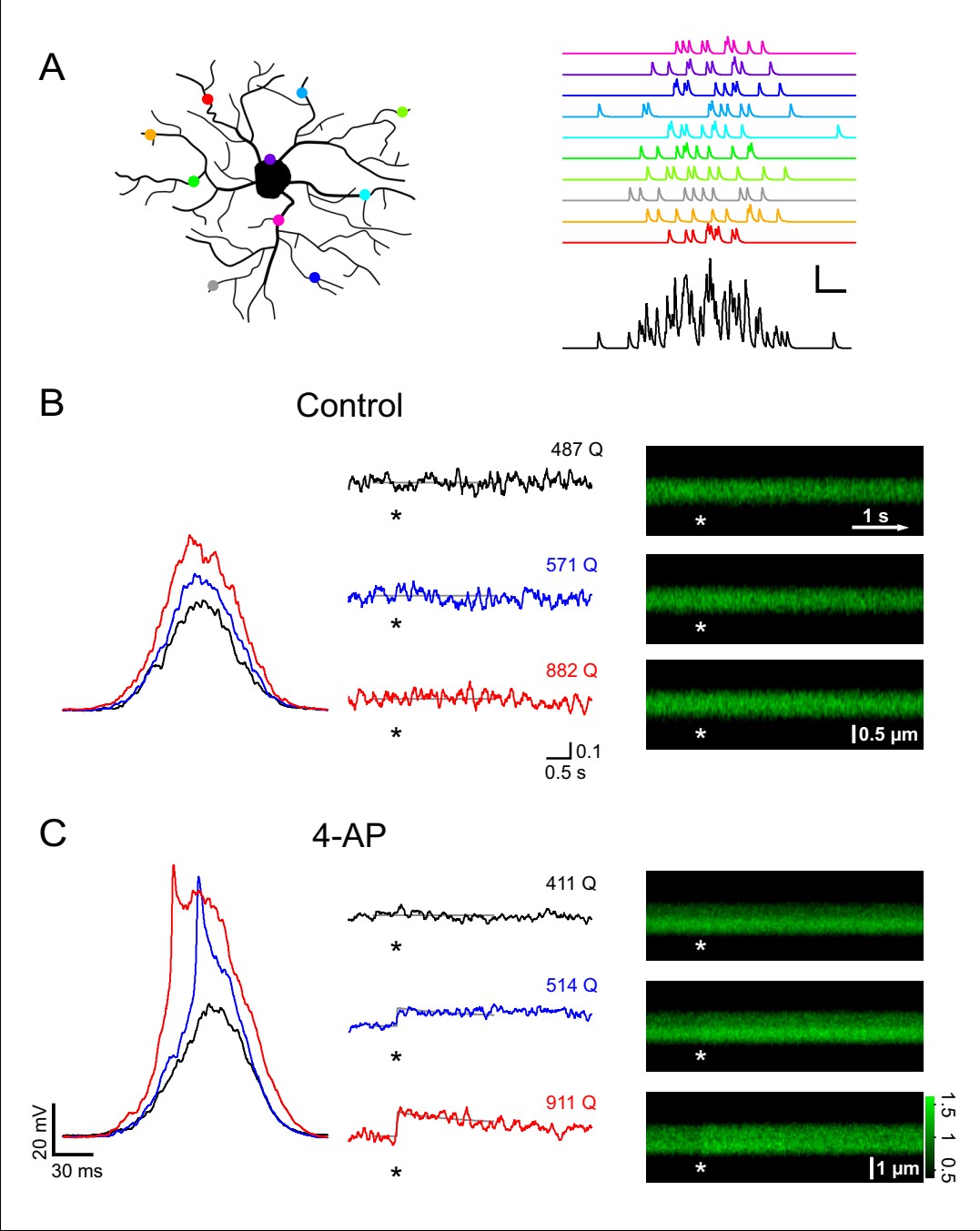

**Figure 6.** A-type channels restrictively gate $Ca^{2+}$ signals that are induced by coincident firing of presynaptic neurons. (**A**) Left: Cartoon illustrating how active synapses (colored dots) may be distributed across the dendritic tree of a NG2 cell. Right: synaptic currents resulting from the synapses color-coded on the left. Each synapse releases 10 vesicles randomly following a Gaussian distribution with a standard deviation of 25 ms. The black trace on the bottom shows the summation of the synaptic currents. Scaling: 20 pA, 20 ms. (**B**) Current clamp recording and 2-photon calcium imaging in the proximal dendrite of a control group NG2 cell. Randomly distributed mock synaptic currents produce a Gaussian-shaped voltage response in the soma (left), which scales with input strength. For each round of stimulation, we drew a new set of random numbers from the Gaussian distribution and generated a new current injection waveform by adding quantal currents at time points assigned by the random numbers. Numbers above the traces indicate the total number of quanta being injected (normalized to a reference $R_{in}$ of 250 MΩ). Note that there is no increase in calcium-indicator fluorescence in the line-scan profiles (middle) or in the line-scans (right) even with the strongest stimulation (indicated by asterisks). Grey lines in the line-scan

*Figure 6 continued on next page*

*Figure 6 continued*

profiles show the fitted template (see Materials and methods). Same scale bars apply to B and C unless noted differently. Unit vertical scale bar: ΔF/F. (C) As in (B) but for a NG2 cell of the 4-AP treated group (4 mM, applied ~10 min before data acquisition). Note the prominent spikelet riding on the two stronger compound synaptic voltage responses, which were never seen without the A-type channel blocker. Further, calcium responses are more readily seen at lower injection strengths. The line-scan on the bottom right shows a very clear response occurring immediately with the stimulation. Grey lines represent fitted templates. Color scale indicates the level of fluorescence in line-scans normalized on the pre-stimulus baseline.

## Discussion

The key finding of the present study is that NG2 cells, which were previously believed to be electrically unexcitable, are effective integrators of synaptic activity. Moreover, they possess two elaborate integration modes governed by A-type K$^+$ channels: local compartmentalized integration in individual dendritic subunits, versus global integration leading to massive Ca$^{2+}$ influx throughout the entire dendritic arborization of NG2 cells.

It has been known for a long time that NG2 cells, formerly also called 'complex astrocytes' (*Steinhauser et al., 1994a*, *1994b*; *Sun and Dietrich, 2013*), express considerable levels of VGCs but their physiological role has not yet been identified. Further, it has been questioned whether the typically small synaptic events in NG2 cells ever recruit VGCs. Earlier studies even proposed that some voltage-gated K$^+$ channels (delayed-rectifier- but not A-type K$^+$ channels) may serve functions other than modifying membrane potential and the authors proposed their involvement in controlling cell cycle timing (*Gallo et al., 1996*; *Knutson et al., 1997*; *Ghiani et al., 1999a*, *1999b*).

Mock PSPs mimicking glutamatergic input, the most widespread neurotransmitter released onto NG2 cells in the CNS, allowed us to systematically study how different types of VGCs shape synaptic input. Given that NG2 cells show voltage-activated Na$^+$ currents (*De Biase et al., 2010*; *Kukley et al., 2010*) and that some reports even found regenerative electrical activity when stimulating NG2 cells with rectangular current injections (*Chittajallu et al., 2004*; *Karadottir et al., 2008*; *Ge et al., 2009*), it could be assumed that these fast channels prominently amplify PSPs. By contrast, we observed that the amplitude of mock PSPs increases linearly with input strength (*Figure 1C*, *Figure 2E*) due to a fine balance of the fast but low amplitude Na$^+$ currents and the much larger but slightly slower A-type K$^+$ currents, with the latter counteracting the amplifying effect of the Na$^+$ channels.

While neurons possess a variety of signaling routes, they primarily integrate synaptic input by summation of electrical signals and generation of an action potential when the summed input crosses the threshold. Single action potentials regularly open VGCCs and are associated with Ca$^{2+}$ transients of constant size. Therefore, intracellular Ca$^{2+}$ levels in neurons are proportional to the number of action potentials in a brief burst or to the frequency of action potentials in a long train (*Helmchen et al., 1996*). NG2 cells also sum incoming electrical synaptic input but, in contrast to neurons and in the absence of action potentials, there is no threshold function for synaptic potentials. The above-described interplay of Na$^+$ and K$^+$ channels allows the size of PSPs to grow proportionally with input strength (*Figure 2E*). Similarly, the amplitude of Ca$^{2+}$ signals in NG2 cells gradually increases with input strength (*Figure 7F*). Thus, in contrast to neurons, NG2 cells use Ca$^{2+}$ levels to gradually encode the level of synaptic activity itself. Both locally induced and global Ca$^{2+}$ signals in NG2 cells decay with a time constant on the order of seconds, while the Ca$^{2+}$ levels in neurons usually return to baseline within several hundred milliseconds. Their low incremental amplitude, their slow return to baseline, and their reliable occurrence even in distal dendrites are characteristics displayed by Ca$^{2+}$ signals in NG2 cells that are key to effectively integrating synaptic input of a wide range of frequencies across their dendritic tree.

Even though the presence of VGCCs in NG2 cells has previously been shown (*Haberlandt et al., 2011*; *Cheli et al., 2014*; *Larson et al., 2016*), their recruitment by synaptic activity has so far remained obscure. Our results demonstrate for the first time that synaptic-like (mock) depolarizations trigger robust and rapid Ca$^{2+}$ signals in the soma and dendrites of NG2 cells. We furthermore provide evidence that the observed Ca$^{2+}$ signals are mediated by low-threshold VGCCs: (1) they were triggered solely by current injection; (2) they started to appear when cells were depolarized

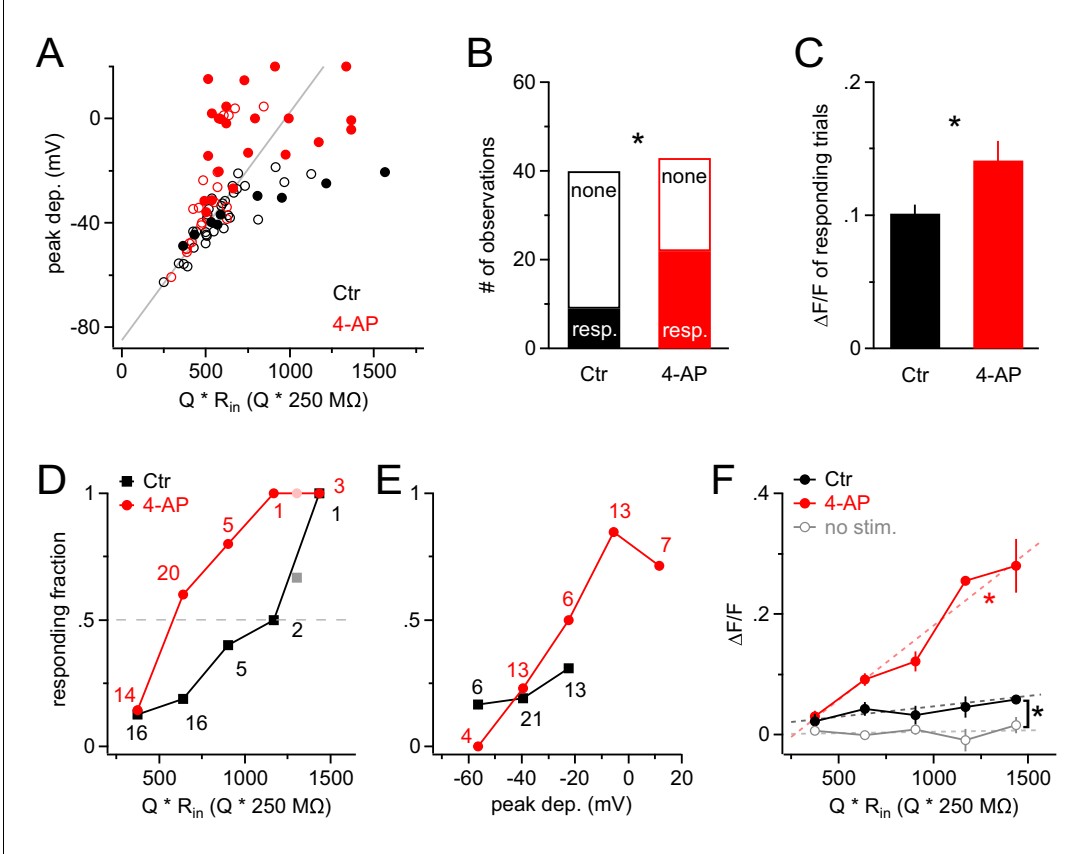

**Figure 7.** A-type K$^+$ channels strongly increase the threshold for synaptic input to generate Ca$^{2+}$ signals in NG2 cells. (**A**) Scatter plot of the peak of the depolarization response to Gaussian-shaped mock PSP injections shown in *Figure 6* versus injection strength for control (black; n=40 trials from nine cells) and 4-AP (red; n=43 trials from seven cells) groups. Peak depolarization is determined as the maximum membrane depolarization reached during a Gaussian-shaped mock PSP. Filled and empty circles indicate trials with and without a Ca$^{2+}$ response, respectively. The grey line indicates how the voltage response size would grow with injection strength in the absence of VGCs. Note that the more strongly NG2 cells are depolarized, the more frequently responding trials are observed, and that 4-AP facilitates the occurrence of large depolarizations at comparable injection strengths (symbols above grey line). The average injection strengths are comparable between the control (615 ± 40.5 Q * R$_{in}$[Q * 250 MΩ]) and 4-AP (640 ± 40.8 Q * R$_{in}$[Q * 250 MΩ]) groups. (**B**) The fraction of responding trials is significantly larger in the 4-AP group. (**C**) Ca$^{2+}$ responses are also significantly larger in 4-AP-treated NG2 cells (Student's *t*-test). Averages are calculated across responding trials only. (**D**) The fraction of Ca$^{2+}$-responding trials increases with injection strength to almost unity in both groups. However, the stimulus–response curve is shifted to the left in the presence of 4-AP, such that at much weaker synaptic input, 50% of dendrites respond with a Ca$^{2+}$ signal (dashed grey line indicates 50% level). Data from individual dendrites have been binned (number of trials per bin indicated by numbers). Lighter color symbols on the right end of the curve denote values resulting from combining trials from the last two bins in each group. (**E**) Data as in (**D**) but plotted against the peak of the depolarization instead of input strength. The control group (black) covers only a small depolarizing voltage range and shows no obvious difference to the responding fraction of dendrites from the 4-AP group, indicating that 4-AP primarily changes the voltage response but not the voltage-dependence of the Ca$^{2+}$ signal. (**F**) The amplitude of the average Ca$^{2+}$ response of the whole population of NG2 cell dendrites increases linearly with the strength of mock synaptic input if A-type K$^+$ channels are blocked (linear correlation test, R$^2$=0.68; dashed red line). Averages were obtained across all trials including non-responding dendritic segments. The amplitude of Ca$^{2+}$ signals in the control group is significantly larger than that in the no-stimulus group (Paired *t*-test). The grey data line represents an average of no-stimulus trials across both control and 4-AP groups.

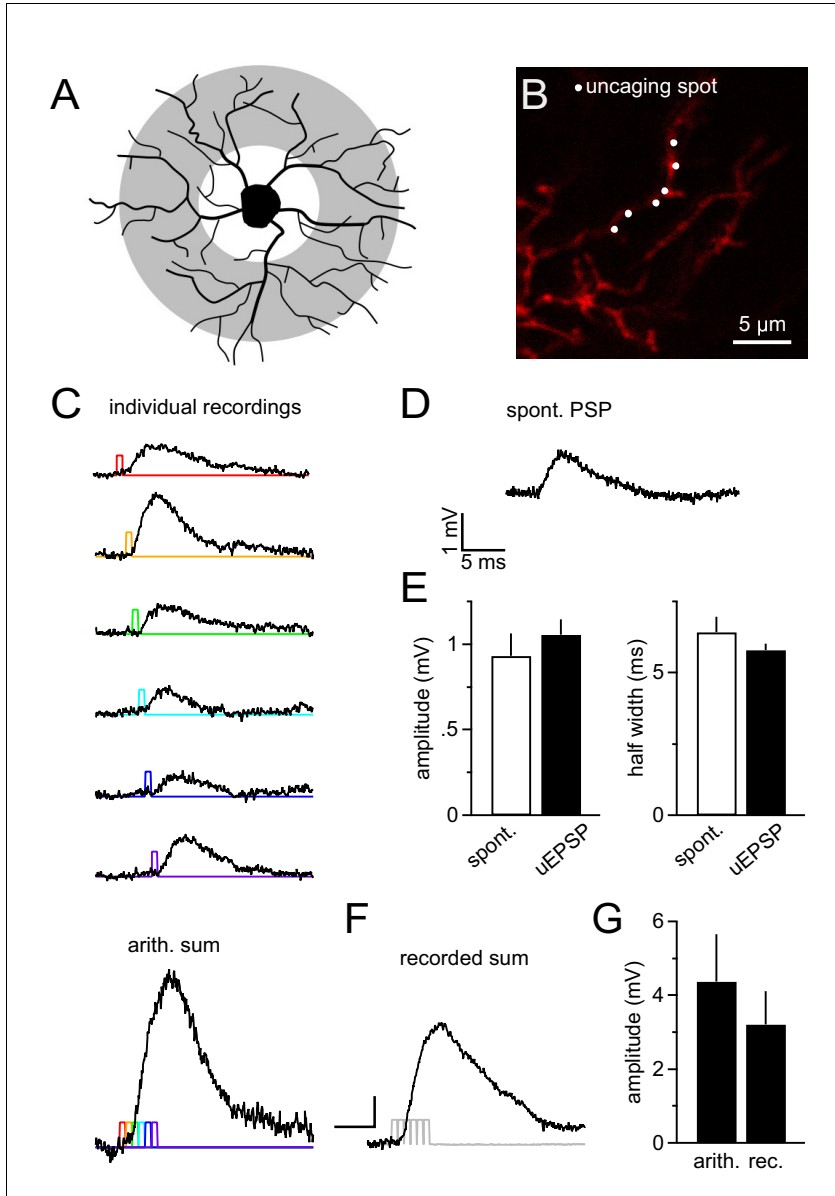

**Figure 8.** NG2 cell dendrites are highly sensitive to local 2-photon glutamate uncaging. (**A**) Grey denotes the target area for 2-photon glutamate uncaging spots on NG2 cells. (**B**) Dye-filled NG2 cell in acute slice scanned with 2-photon microscopy (25 µM Alexa 594). Six uncaging spots (white dots) were placed along a dendritic segment and were stimulated either sequentially (**C**) or nearly instantaneously (**F**). (**C**) Individual voltage responses (uncaging EPSPs, uEPSPs) of the NG2 cell held in current clamp around –85 mV evoked by sequential photo-release of glutamate at the six spots shown in (**B**). Color-coded lines indicate the timing of the laser light pulses. The trace at the bottom represents the arithmetic sum of the individual voltage responses. To make this sum comparable to the 'recorded sum' shown in (**F**), traces were slightly time-shifted (as indicated by the colored laser pulses) to match the unavoidable delay of laser pulses when aiming at near instantaneous uncaging. (**D**) Example recording of a spontaneously occurring PSP in NG2 cells. Note that these PSPs are almost indistinguishable from uEPSPs with respect to their amplitude and kinetics. (**E**) Quantification of amplitude and half width of uEPSP (n=51 of five cells) and spontaneous fast PSPs (n=9 of eight cells). Note the similarity between groups. (**F**) uEPSP evoked by almost simultaneous uncaging at six positions (0.1 ms interval between spots to provide time for moving the laser scanning mirrors). Scaling: 1 mV, 5 ms. (**G**) Arithmetic and recorded sums are very comparable (n=6 dendrites from two cells, Paired *t*-test). To correct for any time-dependent run-down (found to be ~10%), the sequence of individual responses was acquired before and after the simultaneous six-pulse uncaging and the bar graph depicts the average of both.

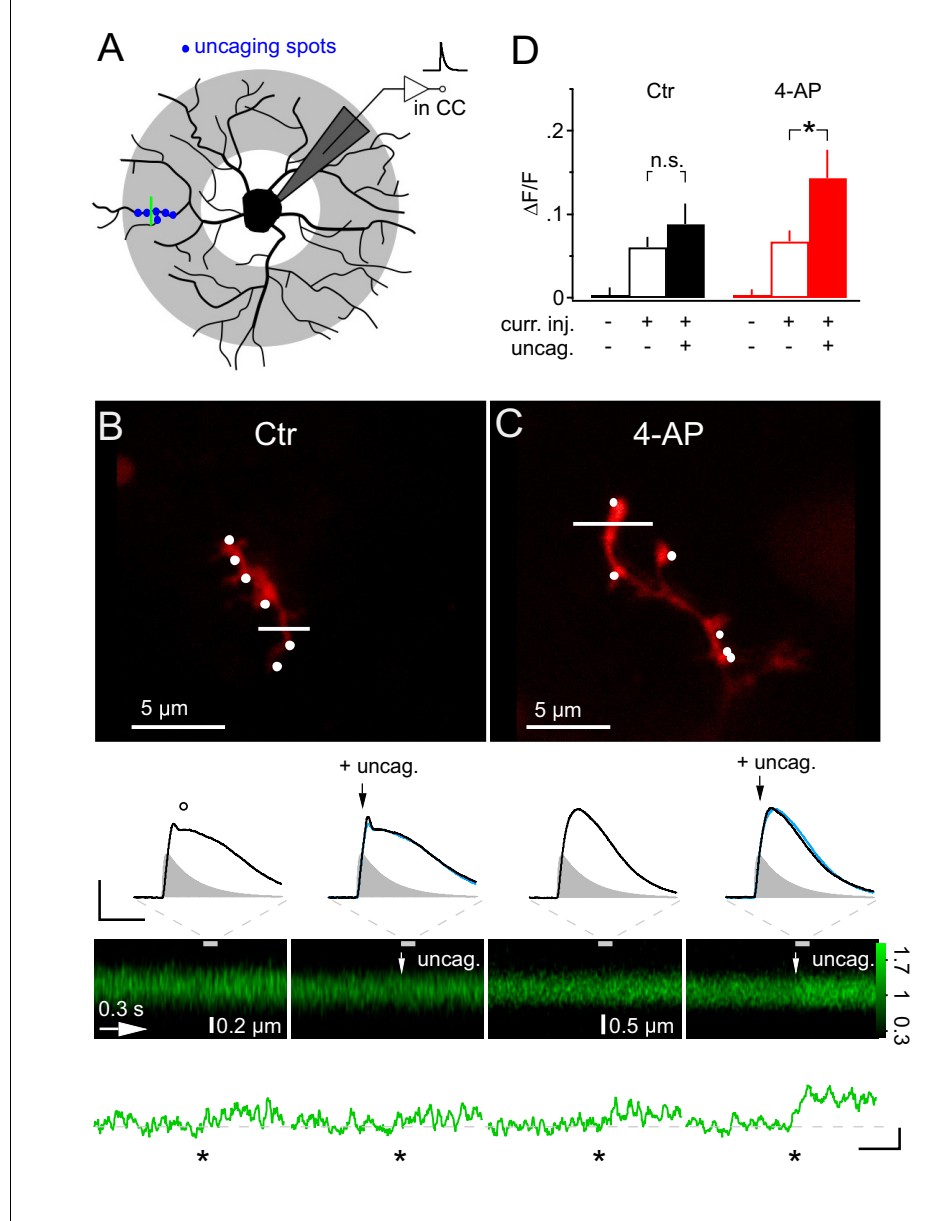

**Figure 9.** Local glutamate uncaging triggers 4-AP-dependent rapid $Ca^{2+}$ signaling in NG2 cells dendrites. (**A**) NG2 cells were patch-clamped and stimulated with mock PSPs in current clamp mode. Dendritic target area for uncaging and imaging is shaded in grey. In this experiment, we used a quantal IPSC waveform to generate the mock PSP. The green line indicates strategy for selecting line-scans, typically within the region of the uncaging spots (blue dots). (**B**) Adding glutamate uncaging to mock PSP stimulation does not alter the $Ca^{2+}$ response in control conditions. Frame-scan shows the position of the six uncaging spots (white dots, 0.1 ms or 1 ms interval) and the line-scan used to quantify any $Ca^{2+}$ response of Fluo-4 (white line). Second row, left: The voltage response is hardly increased when simultaneous glutamate uncaging at the six indicated positions is added to the mock PSP current injection. Note the small dip (circle) after the peak of the voltage response, which is caused by activation of A-type $K^+$ channels. Right: The first uncaging pulse is added 5 ms after the start of the current injection, denoted by black arrows. The blue line indicates the control mock PSP for comparison (scaling: 20 mV, 40 ms). Injected current waveform derived from miniature IPSCs are shown as grey shaded areas. Note that the voltage responses are brief compared to the $Ca^{2+}$ signal and are shown on an expanded time scale indicated by the grey dashed lines. Third and fourth row: The line-scans (the grey bar indicates the interval during which the voltage responses have been acquired) and corresponding spatially averaged profiles (green traces on bottom) do not show visually identifiable responses. However, note that the $Ca^{2+}$ increases generated by mock PSP injection alone and by combined application of mock PSP and local glutamate uncaging (indicated by white arrow) were

*Figure 9 continued on next page*

*Figure 9 continued*

not significantly different (asterisk indicates time of current injection). (**C**) When A-type K$^+$ channels are inactive, local glutamate uncaging significantly boosts dendritic Ca$^{2+}$ signal. Panels as in B but for an NG2 cell from the 4-AP group. Note that in the presence of 4-AP, glutamate uncaging (black and white arrows) hardly increases the voltage response when compared to the response to mock PSP injection alone (left). The small dip seen in control conditions after the peak of the depolarization is absent. The line-scans and the line-scan profiles show a clear Ca$^{2+}$ response when mock PSP and six spots of glutamate uncaging are co-applied. Scaling: 0.2 ΔF/F, 0.3 s. (**D**) Increase in fluorescence of the calcium indicator dye Fluo-4 is significantly boosted by additional uncaging if A-type K$^+$ channels are inhibited. ΔF/F values were calculated from periods before and soon after (100 ms) the stimulation or during baseline period with no stimulation of line-scan profiles shown in (**B**) and (**C**). Control: n=26 dendrites of 10 cells; 4-AP: n=30 dendrites of 16 cells.

beyond −50 to −40 mV; (3) they were blocked by nickel/cadmium and by a cocktail of the R- and T-type channel blockers SNX-482 and TTA-P2; (4) they were potentiated by blocking K$^+$ channels and showed a rapid rise, implying that the Ca$^{2+}$ source was active for only a few milliseconds; (5) they were resistant to blocking Na$^+$/Ca$^{2+}$ exchangers or Ca$^{2+}$ stores by 10 μM KB-R7943 and thapsigargin, respectively. Thus, the Ca$^{2+}$ signals we recorded in NG2 cells are quite distinct from the store-mediated Ca$^{2+}$ elevations seen in astrocytes, which occur and decay over many seconds. Haberlandt et al. reported that depolarization-induced Ca$^{2+}$ signals in NG2 cells are very sensitive to thapsigargin, seemingly contradicting our findings (*Haberlandt et al., 2011*). However, the aforementioned study used a much stronger stimulus and depolarized NG2 cells for 100 ms to +20 mV (voltage clamp), and the resulting Ca$^{2+}$ signal showed kinetics an order of magnitude slower than those reported here. Thus, it is conceivable that NG2 cells generally can generate store-dependent Ca$^{2+}$ signals but that store-dependent signaling is not readily recruited by synaptic activity studied here.

The kinetics and magnitude of Ca$^{2+}$ signals in NG2 cell dendrites, induced by local synaptic stimulation or local glutamate uncaging, match those evoked by somatic current injection very well, suggesting that they are also caused by recruitment of VGCCs. Under the conditions used here, mimicking the activity of a very small number of synapses, Ca$^{2+}$ entry through glutamate receptors appears not to be involved because 4-AP boosted the response and because executing the glutamate uncaging protocol alone never triggered a Ca$^{2+}$ response (unpublished observations). However, this does not imply that glutamate receptors cannot be a source of Ca$^{2+}$ entry if more widespread opening of receptors is achieved (see *Ge et al., 2006*).

Ca$^{2+}$ signals in NG2 cells are restrictively gated by A-type K$^+$ channels. This may be one of the reasons why previous publications concluded that synaptically driven Ca$^{2+}$ signals in NG2 cells are unlikely to occur (*Velez-Fort et al., 2010*; *Haberlandt et al., 2011*). By which mechanisms do A-type K$^+$ channels gate Ca$^{2+}$ signals? Since the voltage-dependence of the Ca$^{2+}$ signal itself remained unchanged after blocking A-type K$^+$ channels (*Figure 7E*), it is very likely that A-type channels under control conditions render synaptic input less effective in causing Ca$^{2+}$ signals by shortening and decreasing the associated synaptic depolarization. In consequence, the impact of synaptic transmission on Ca$^{2+}$ signaling in NG2 cells is dependent on the activity of A-type K$^+$ channels. Considering that Ca$^{2+}$ levels have been implicated in the developmental behavior and gene expression of NG2 cells, A-type K$^+$ channels are likely to play a key regulatory role in whether and how neuronal transmitter release affects oligodendrogenesis and myelination (*Pende et al., 1994*; *Paez et al., 2009*, *2010*). The fact that NG2 cells are most responsive to synaptic input from neurons when A-type channels are suppressed raises the question of under which conditions such suppression is achieved. Activity of A-type channels can temporarily be reduced when depolarizations bring them into a state of inactivation. Thus, those patterns of synaptic input that maximize inactivation of A-type channels enhance the impact of axonal activity on NG2 cell behavior. Based on previous reports on the kinetics of A-type K$^+$ channels (*Steinhauser et al., 1994b*), our data allows us to predict that brief trains of action potentials (50–100 ms) should be very effective at inactivating A-type channels and will open the 'calcium-gate' in NG2 cells. Apart from functionally and temporarily reducing A-type currents by inactivation, transcriptional regulation may be another way to control A-type current amplitude in NG2 cells. Indeed, there is evidence for developmental and reactive

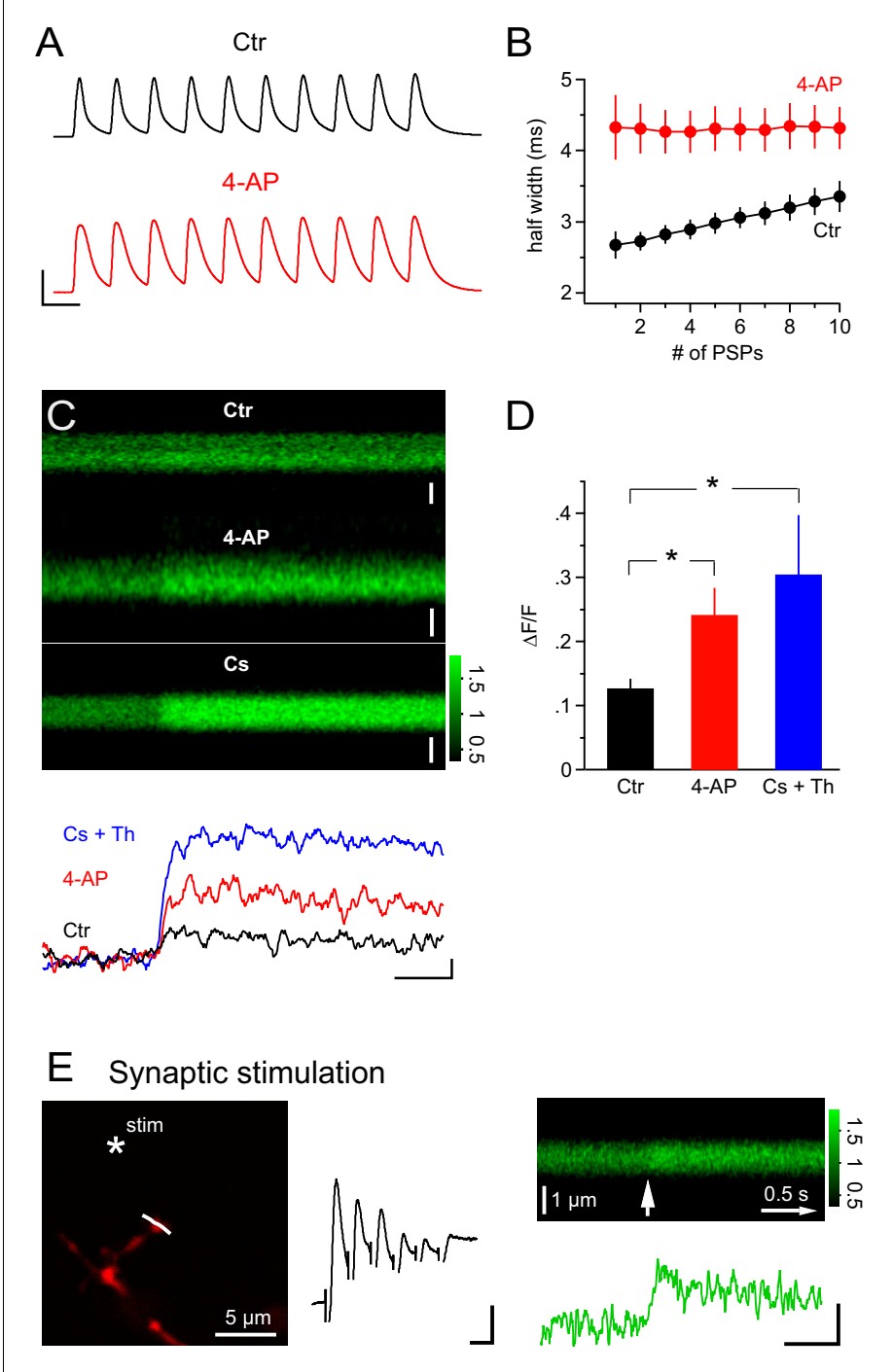

**Figure 10.** PSPs inactivate A-type K$^+$ channels and synaptic stimulation induces Ca$^{2+}$ transients independent of Ca$^{2+}$ stores. (**A**) Example recordings of trains of mock PSPs (100 Hz, 100 ms) in control and 4-AP groups. Note that the half-width of PSPs in the control group progressively increases during the train. By contrast, PSPs in the 4-AP group are broader in the beginning and their half-width does not increase during the train. Scaling: 50 mV, 10 ms. (**B**) Activity-dependent broadening of PSPs in a train of the control group. Average half-width of PSPs (n=7) is smaller in the control group and approaches that of PSPs in the 4-AP group (n=8) during a 10 stimuli train. (**C**) 2-photon line-scans (top) and corresponding profiles (bottom) of Fluo-4 fluorescence in dendrites of NG2 cells during injection of mock PSPs in control, 4-AP and Cs$^+$ groups. Note the more pronounced Ca$^{2+}$ signal in the presence of 4-AP, which is probably due to the increased width of PSPs enhancing Ca$^{2+}$ tail currents. Scaling: 1 µm (control), 0.5 µm (4-AP) and 1 µm (Cs$^+$) for line-scans; 0.1 ΔF/F, 0.5 s for line-scan profiles. All line-scans and

*Figure 10 continued on next page*

*Figure 10 continued*

line-scan profiles shown are the average of three repetitions. Note that 1 µM thapsigargin was included in the pipette solution for the Cs$^+$ group. (D) The average ΔF/F induced by a 100 Hz 100 ms train of PSPs is significantly increased in the 4-AP group. (Student's *t*-test: control vs 4-AP group). Blocking K$^+$ conductance by using Cs-gluconate-based internal solution also significantly increases the ΔF/F induced by train PSPs (Wilcoxon-Mann-Whitney test: control vs Cs$^+$ group, see Materials and methods). (E) Left: Frame-scan (Alexa 594) shows the position of the stimulating electrode (asterisk) and the line-scan (white line) used to quantify the fluorescence response of the Ca$^{2+}$ indicator dye Fluo-4 (200 µM). Middle: Electrical response measured at the soma of the NG2 cell held in current clamp mode. Note the small amplitude of the response, which typically ranged between 5.1 and 10.1 mV (5 or 6 stimuli at 100Hz). Scaling: 2 mV, 10 ms. Right top: Average line-scan of Fluo-4 signal from three repetitions of local synaptic stimulation at the location shown in (A). The white arrow indicates the time of the first of six synaptic stimulations. Right bottom: Synaptically induced Ca$^{2+}$ signal calculated from line-scan showed similar kinetics and amplitude as calcium signals evoked by current injection or glutamate uncaging. The presence of 1 µM thapsigargin in the pipette solution excludes the potential contribution of internal calcium stores to the Ca$^{2+}$ response. Scaling: 0.2 ΔF/F, 0.5 s.

regulation of K$^+$ channels in response to ischemia/injury (*Lytle et al., 2009*; *Pivonkova et al., 2010*). Such regulated expression of A-type K$^+$ channels may allow NG2 cells to permanently tune their responsiveness to incoming neuronal activity. This adaptation is probably required to match the responsiveness of NG2 cells to patterns of activity, which vary across regional neuronal circuits throughout the brain (*Sun and Dietrich, 2013*).

When performing glutamate uncaging along NG2 cell dendrites, we regularly observed depolarizations with similar size and kinetics to miniature EPSPs. For the first time, this directly demonstrates that there are many synapse-like clusters of glutamate receptors along many dendrites of NG2 cells. Such widespread responsiveness to glutamate throughout the dendritic tree indicates that NG2 cells may be capable of dendritic computation. Dendritic computation in NG2 cells is further suggested by the fact that local glutamate uncaging boosted the dendritic Ca$^{2+}$ response despite the small amplitude of the associated somatic voltage responses. In particular, the tiny incremental depolarization caused by adding dendritic uncaging (Δ 2.5 mV) is far too small to significantly enhance the recruitment of VGCCs at the soma (*Figure 9C*). This implies that the dendritic voltage excursion caused by glutamate uncaging is actually substantially larger than that measured at the soma, and that this more pronounced local depolarization recruits Ca$^{2+}$ channels in the dendrite. Considering the thin diameter of NG2 cell dendrites, such severe attenuation of dendritic voltage is not surprising (*Chan et al., 2013*; *Sun and Dietrich, 2013*). A larger dendritic voltage excursion is also suggested by the level of the absolute depolarization of the soma (as opposed to the uncaging-induced increment in depolarization considered above), which is clearly below the voltage range typically required to reliably induce Ca$^{2+}$ signals (*Figure 7*). Further, all Ca$^{2+}$ signals recorded in dendrites – regardless of whether induced by synaptic stimulation, by a train of mock PSPs, by gaussian distributed mock PSPs, or by glutamate uncaging – display a rapid rise, implying that the Ca$^{2+}$ source must be very close to the optical recording site, the dendrite. Finally, Ca$^{2+}$ signals induced in the soma rapidly spread into all dendrites and reached even the distal ends with undiminished amplitudes. This can only be achieved if VGCCs are present all along the dendrites to regenerate the signal. As we did not detect any dendrites that did not generate Ca$^{2+}$ responses, the data indicate that most, if not all, dendrites of NG2 cells are equipped with VGCCs.

There is a similar line of argument for a dendritic localization of A-type channels in NG2 cells. Blocking these channels with 4-AP allowed glutamate uncaging to boost the Ca$^{2+}$ response, i.e. to recruit more Ca$^{2+}$ channels by a stronger depolarization compared to control conditions. However, the resulting depolarization measured at the soma was negligibly increased by the K$^+$ channel blocker when compared to control conditions and thus cannot explain additional recruitment of Ca$^{2+}$ channels (*Figure 9*). Therefore, application of 4-AP must have allowed for a larger depolarization of the local membrane in the dendrite in response to uncaging, which was not visible in the somatic recording. The most parsimonious explanation is that 4-AP was also acting on dendritic A-type K$^+$ channels, which renders dendritic glutamate uncaging more effective in generating a larger local depolarization and recruiting dendritic VGCCs. Taken together, our data show that NG2

cells express voltage-dependent $Ca^{2+}$ and $K^+$ channels in their dendrites and that these channels actively shape synaptic input as independent processing units.

It has been put forward that synaptic transmission from unmyelinated axons to NG2 cells may play a pivotal role in triggering these responses of NG2 cells (*Wake et al., 2011*; *Hines et al., 2015*; *Mensch et al., 2015*), also reviewed in (*Almeida and Lyons, 2014*; *Petersen and Monk, 2015*). Based on our present findings, we propose that synaptically induced $Ca^{2+}$ signaling is well-suited to initiate actions of NG2 cells in the context of activity-dependent myelination. First, intracellular $Ca^{2+}$ levels in NG2 cells allow for optimal temporal and spatial summation of incoming neuronal input and their dependence on the functional state of A-type channels provides certain types of plasticity for synchronized activity. Second, NG2 cells seem to generate $Ca^{2+}$ signals in at least two different spatial domains, which may serve distinct roles in the context of activity-dependent myelination. Dendritic input, possibly even induced by an individual axon, can trigger local $Ca^{2+}$ signals. Such local $Ca^{2+}$ signals could be involved in stabilizing or enhancing contact sites with individual axons (*Hines et al., 2015*; *Mensch et al., 2015*) and could enable axon-specific integration and computation. On the other hand, global $Ca^{2+}$ signals, which are induced by a larger number of active synapses throughout the dendritic tree, spread within the NG2 cell and involve the soma, are particularly qualified for triggering global cellular actions, such as division, differentiation, survival or motility of NG2 cells in response to neuronal activity. Further, our finding of local processing of synaptic input raises the question of how local dendritic computation in NG2 cells is relevant for myelination and whether the spatial arrangement of synaptic junctions on NG2 cells affects myelination of the connected axon.

In conclusion, our study shows for the first time that NG2 cells possess a dedicated synaptic integration repertoire, distinct from that of neurons and other types of glial cells, by encoding the number of active synapses in a graded fashion. This integration builds on $Ca^{2+}$ signals in different spatial domains and appears as a specialized type of $Ca^{2+}$ excitability of NG2 cells. Voltage-gated $K^+$ channels dampen $Ca^{2+}$ signals but this can be overcome by temporally and spatially coincident synaptic depolarizations. Thus, specific patterns of axonal activity may promote myelination by instructing NG2 cells to interact with axons or generate new oligodendrocytes.

## Materials and methods

### Slice preparation

Post-natal 7–15-day old (male and female) transgenic mice expressing DsRed fluorescent protein under control of the promoter of mouse *Cspg4* gene were used in this study (NG2-DsRed transgenic mouse line [*Zhu et al., 2008*]). After being anesthetized with isoflurane, the mouse brain was removed from the skull rapidly and submerged into ice-cold dissecting solution containing (in mM): 87 NaCl, 2.5 KCl, 1.25 $NaH_2PO_4$, 7 $MgCl_2$, 0.5 $CaCl_2$, 25 $NaHCO_3$, 25 glucose, and 75 sucrose (gassed with 95% $O_2$/5% $CO_2$). Frontal hippocampal slices (300 μm) were prepared on a vibratome (Leica VT 1200S). The slices were then quickly transferred to a submerged chamber containing dissecting solution at 35°C for 25 min before being stored at room temperature in (ACSF, mM): 124 NaCl, 3 KCl, 1.25 $NaH_2PO_4$, 2 $MgCl_2$, 2 $CaCl_2$, 26 $NaHCO_3$, 10 glucose (gassed with 95% $O_2$/5% $CO_2$). Electrophysiological recording was started not earlier than 1h after dissection.

### Patch-clamp recordings

Whole-cell patch-clamp recordings were obtained from $DsRed^+$ NG2 cells in hippocampal CA1 Stratum Radiatum region. Cells were held in current-clamp mode at $-85$ mV (NPI SEC-05, Dagan BVC-700A or HEKA EPC10 amplifier) while continuously perfusing slices with ACSF at room temperature. For experiments applying $Cd^{2+}$ and $Ni^{2+}$, $NaH_2PO_4$ was removed from ACSF to avoid precipitation. Patch pipettes were pulled using a vertical puller (Model PP-830, Narishige) with a resistance of 4.5–5.5 MΩ. Pipette solution contained (in mM): 125 K-gluconate, 4 $Na_2$-ATP, 2 $MgCl_2$, 10 HEPES, 20 KCl, 3 NaCl, 0.5 EGTA, 0.1% Lucifer Yellow (pH=7.3, 280–290 mOsm). Cs-gluconate-based pipette solution contained (in mM): 150 Cs-gluconate, 2 $MgCl_2$, 15 CsCl, 2 $Na_2$ATP, 10 HEPES, 1 μM thapsigargin (pH=7.3, 280–290 mOsm). For $Ca^{2+}$-imaging experiments, EGTA and Lucifer Yellow were excluded from the pipette solution, and replaced by 200 μM $Ca^{2+}$ indicator Fluo-4 (Thermo Fisher) plus either 25 μM Alexa Fluor 594 (Thermo Fisher) or 100 μM tetramethylrhodamine-biocytin (TMR;

Thermo Fisher). The liquid junction potential was corrected by adjusting the zero-current position to $-10$ mV before the sealing procedure. We used various software packages, including pClamp10 (Molecular Devices), PATCHMASTER (HEKA), WinWCP (Strathclyde Electrophysiology Software, University of Strathclyde Glasgow) or Igor Pro software (WaveMetrics, recording, analysis and figure preparation) running mafPC (courtesy of M. A. Xu-Friedman). The responses were recorded with a sampling rate of 20 kHz (DigiData 1440 from Molecular Devices, or NI USB-6229 from National Instruments) and were low-pass filtered at 3 or 10 kHz. The average input resistance of all NG2 cells included in the study was $251.4 \pm 18.4$ M$\Omega$ (n=132, range 37 to 1483 M$\Omega$). The average resting potential (no current injection) was $-83.8 \pm 0.7$ mV (n=132).

In order to evoke a maximal Schaffer-collateral mediated PSP, a mono-polar or bi-polar stimulating electrode was placed in the CA1 Stratum Radiatum region approximately 20–30 μm from the NG2 cell soma (isolated stimulator, A-M systems Model 2100, 0.2 to 0.5 ms). Starting from low values, stimulation intensity was gradually increased until a maximal response was reached. For local synaptic stimulation, a mono-polar stimulating electrode was placed close to the target dendritic segment (~5 μm in distance). A train of 5–6 stimuli (0.1 ms) at 100 Hz was applied during 2-photon $Ca^{2+}$ imaging of the target dendritic segment. To remove any endogenous suppression of transmitter release by ambient adenosine, 1 μM 8-cyclopentyl-1,3-dipropylxanthine (DPCPX) was added to the perfusion medium during all synaptic stimulation experiments. 1 μM thapsigargin was added to pipette solution for local synaptic stimulation experiments.

For mock PSP recordings, the injected current waveform template was derived from previously recorded miniature EPSCs in NG2 cells. The amplitude of this current, the quantal amplitude, was set to 12 pA according to the range of values reported for miniature EPSCs previously (*Bergles et al., 2000*; *Lin et al., 2005*; *Kukley et al., 2010*; *Chan et al., 2013*; *Passlick et al., 2016*). To achieve a stronger stimulation, the current waveform was multiplied with an integer number, Q, representing the number of quanta contained in the stimulus. The electrical impact of a synapse depends on the individual passive membrane properties of the NG2 cell, membrane resistance, $R_{in}$, and membrane capacitance. Because membrane resistance, but not capacitance, varies greatly among NG2 cells (*Kukley et al., 2010*) and the resistance has a larger impact on the voltage response, we calculate the strength of our mock current injection as the product $Q * R_{in}$. To facilitate interpretation of this product, i.e. to determine the quantal content, we express it in multiples of 250 M$\Omega$, the population average input resistance of NG2 cells. For example, injecting 100 Q into $R_{in}$=250 M$\Omega$ yields $Q * R_{in}$=25,000 (Q * M$\Omega$) and injecting 50 Q into $R_{in}$=500 M$\Omega$ yields the same product, $Q * R_{in}$=25,000 (Q * M$\Omega$). Therefore, the stimulations are equally effective and both would be expressed as 100 $Q * R_{in}$ (Q * 250 M$\Omega$). For simplicity, we neglected changes in Q due to varying driving force. We also injected a small inverted, hyperpolarizing EPSC waveform to produce a purely passive voltage deflection in each recording. The average hyperpolarizing response size was $-6.4 \pm 0.2$ mV (54 cells; *Figure 1C* and *Figure 2*). To comparatively illustrate the amplitude and waveform of a voltage response to an arbitrarily-sized mock current injection not affected by voltage-gated channels, we multiplied the small passive response obtained in the same cell by the (negative) amplitude ratio of the arbitrary injection over the small inverted EPSC injection.

To rule out the possibility that increased osmolarity of the ACSF due to the addition of TEA changes mock PSPs (*Figure 2A and B*), we ran a set of control experiments with increased osmolarity by adding sucrose (20 mM), which did not affect mock PSPs (n=5, Paired *t*-test).

## Immunohistochemistry

After recording, slices were fixed and stained for NG2 as previously described (*Kukley et al., 2007*; *Kukley et al., 2010*). A total of 12 recorded DsRed$^+$ cells were tested and verified to be NG2 positive.

## 2-photon $Ca^{2+}$ imaging

2-photon $Ca^{2+}$ imaging experiments were performed either with a Prairie Technologies Ultima Multiphoton Microscopy System (Bruker) or with a Nikon A1R MP 2-photon scanning microscopy (Nikon), or with a Scientifica SliceScope 2-photon microscope system (Scientifica). The imaging laser's wavelength was set at 820 nm and fluorescence was collected by a 60X objective (NA 1.0; Nikon), a 25X objective (NA 1.10; Nikon), or a 20X objective (NA 1.0; Olympus). Green (Fluo-4, 200 μM) and red

(Alexa594 25 µM or TMR 100 µM) fluorescence were collected via 490–560/575/585–630 nm or 500–550/560/570–655 nm, or 500–550/565/590–650 nm filter cubes. The pre-chirped imaging laser (Chameleon Vision II, Coherent) intensity was ~9–12 mW (Prairie), ~8 mW (Nikon) and ~7 mW (Scientifica) at the surface of the specimen. $Ca^{2+}$ signals in line-scan and frame-scan mode were acquired at a frequency of 100–300 Hz and 30 Hz (resonant scanner), respectively. Line-scan profiles illustrated in the figures were smoothed using a sliding average of five (*Figures 3* and *10E*) or nine points (*Figures 4*, *6*, *9* and *10C*). Pictured line-scan images were sequentially filtered with two orthogonal 1-D gaussian filters (width of 2 pixel in time, horizontal and 1 pixel in space, vertical, *Figures 3*, *6*, *9* and *10*). The ΔF/F frame-scan shown in *Figure 5B* (right) was filtered with a 2-D gaussian filter (width=2 pixels).

## 2-photon glutamate uncaging

2-photon glutamate uncaging experiments were performed with Prairie Technologies Ultima Multiphoton Microscopy System (Bruker) equipped with two tunable Ti:sapphire lasers and two scan heads. The uncaging laser's wavelength was set to 720 nm. The hippocampal slice was bathed with 5 mM MNI-glutamate (Tocris Bioscience) and the perfusion was stopped for up to 30 min. Stability of resting membrane potential, morphology and resting $Ca^{2+}$ concentration (baseline fluorescence) were used to rule out damage to the cell potentially caused by stopping the perfusion. The pointing function of the second scan head was used to direct the uncaging laser to the target locations through the 'Triggersync' control software (Bruker). The software also controlled application of a brief laser pulse to uncage the MNI-glutamate (0.65 ms, Pockels cell modulation, intensity of ~35–45 mW at the surface of the specimen).

## Analysis of line- and frame-scans

Line-scans were background-subtracted and the fluorescence of dendrites or somata was spatially averaged within each individual line-scan to obtain a single mean fluorescence value per time/line scan, the line-scan profile. Background was estimated from line-scans obtained with the laser off, which for 2-photon microscopy, provides a close to optimum approximation of the background signal at the position of the scanned structure. Fluorescence responses (ΔF/F) in line-scan profiles were quantified as $(Fpeak-F_0)/F_0$, where $F_{peak}$ was estimated by the mean fluorescence value during a 150 ms interval starting 100 ms after the beginning of the current injection, and $F_0$ by an average of the fluorescence during a 150 ms baseline period before the current injection. Calcium responses in frame-scans (*Figure 5*) were quantified as ΔG/R ratios per region of interest (ROI). ΔG was calculated for each ROI from the Fluo-4 fluorescence as $G_{peak}-G_0$, where $G_{peak}$ and $G_0$ represent mean fluorescence values per ROI obtained from two series of successive frames after and before the current stimulus, respectively, as indicated in *Figure 5A*. R was calculated for each ROI from a background-corrected average frame calculated across all frames used for $G_{peak}$ and $G_0$ analysis.

## Quantification of decay time constants from line-scan profiles

The decay time constants ($\tau_{decay}$) were calculated by fitting a long line-scan profile (at least 7 s in duration) with either a mono- or a bi-exponential function when indicated by F-statistics. The reported mean $\tau_{decay}$ is the average of the $\tau_{decay}$ from two cells showing a mono-exponential decay and the weighted $\tau_{decay}$ from seven cells requiring a bi-exponential fit.

## Template-based detection of $Ca^{2+}$ responses

A $Ca^{2+}$ response template was derived from our recordings that showed clear $Ca^{2+}$ responses (length 2.5 s). The template was fitted to the fluorescence line-scan profile and the detection criterion was calculated according to *Clements and Bekkers (1997)*. Trials were classified as responses if their detection criterion exceeded 3.0. This threshold was previously reported to yield an optimum between sensitivity and selectivity (*Clements and Bekkers, 1997*), and in our hands, selected traces were also easily classified as responses by human eye.

To determine the onset of the $Ca^{2+}$ signals in dendrites of NG2 cells from the time-lapse frame-scans, we first calculated brightness-over-time profiles from ROIs as described above (see *Figure 5D*: for the analysis, profiles were re-sampled at 1 kHz by linear interpolation). The template (length 1 s, derived from the average brightness-over-time profiles across all ROIs) was slid

along the profiles and at each position scaled to achieve an optimal match. The detection criterion was calculated as above at each position and the onset of response (same detection threshold as above) was read from the position of the template yielding the best detection value. These analysis routines were coded in the Igor Pro software.

## Predicting diffusional propagation of Ca²⁺ signals into dendrites

We previously derived equations to approximate the speed of propagation of a Ca$^{2+}$ signal in dendrites of neurons (*Matthews et al., 2013*) based on the apparent diffusion coefficient of Ca$^{2+}$ in the cytosol, $D_{app}$, and the time of Ca$^{2+}$ extrusion. $D_{app}$ was calculated according to *Zhou and Neher (1993)* and *Wagner and Keizer (1994)*, assuming that NG2 cells only contain Fluo-4 (200 µM, $K_d$=335 nM, D=100 µm$^2$/s [*Gabso et al., 1997*]) and immobile endogenous calcium buffer (with $\kappa_f$=20) as the only relevant calcium-binding species. Further, we assume [Ca]$_{rest}$=100 nM and $D_{Ca}$=223 µm$^2$/s (*Allbritton et al., 1992*) and use $\tau_{decay}$ of our Ca$^{2+}$ signals (3.16 s). Based on these values, we calculated the expected purely diffusional propagation of a Ca$^{2+}$ signal in NG2 cell dendrites according to $\sqrt{2D_{app}}\sqrt{t}$ and $2t\sqrt{\frac{D_{app}}{\tau_{decay}}}$ (*Matthews et al., 2013*) for the first 316 ms and thereafter, respectively (*Figure 5F*).

## Statistics

Data are expressed as mean ± SEM. The Cs$^+$ group in *Figure 10C* was added during a manuscript revision period and did not pass the normality test (Shapiro-Wilk test). Therefore we ran a separate non-parametrical test to compare it to the control group (Wilcoxon-Mann-Whitney test). The significance level was set at alpha=0.05 for all tests. All statistical tests were two-sided.

## Acknowledgements

This study was supported by Deutsche Forschungsgemeinschaft (SPP 1757, SFB1089, DI853/3 and INST 1172/15-1 to DD, and SFB 1089, SCHO820/5 to SS), Bundesministerium für Bildung und Forschung (01GQ0806 to SS) and BONFOR grant (to WS). We thank Stefan Remy for his valuable input on the manuscript and Chris Bolychevsky for correcting the text. We are grateful to Pia Stausberg, Stefanie Lennartz and Julia Enders for their excellent technical assistance.

## Additional information

### Funding

| Funder | Grant reference number | Author |
| --- | --- | --- |
| BONFOR | | Wenjing Sun |
| Deutsche Forschungsgemeinschaft | SFB 1089 | Susanne Schoch |
| Bundesministerium für Bildung und Forschung | 01GQ0806 | Susanne Schoch |
| Deutsche Forschungsgemeinschaft | SPP 1757 | Dirk Dietrich |
| Deutsche Forschungsgemeinschaft | SFB 1089 | Dirk Dietrich |
| Deutsche Forschungsgemeinschaft | DI853/3 | Dirk Dietrich |
| Deutsche Forschungsgemeinschaft | INST 1172/15-1 | Dirk Dietrich |
| Deutsche Forschungsgemeinschaft | SCHO820/5 | Susanne Schoch |

The funders had no role in study design, data collection and interpretation, or the decision to submit the work for publication.

## Author contributions
WS, Conception and design, Acquisition of data, Analysis and interpretation of data, Drafting or revising the article; EAM, Acquisition of data, Analysis and interpretation of data, Drafting or revising the article; VN, Acquisition of data, Analysis and interpretation of data; SS, Drafting or revising the article; DD, Conception and design, Analysis and interpretation of data, Drafting or revising the article

## Author ORCIDs
Wenjing Sun, http://orcid.org/0000-0002-0905-7420
Dirk Dietrich, http://orcid.org/0000-0002-4307-2448

## Ethics
Animal experimentation: This study was performed in accordance with national and institutional guidelines for the care and use of laboratory animals. Every effort was made to minimize suffering.

## Additional files

### Supplementary files
• Supplementary file 1. The information of all statistical tests used in the text or figures, including the sample size, the names of the statistical tests, exact P values and additional information.

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
