## [Decision Letter]

[Editors’ note: this article was originally rejected after discussions between the reviewers, but the authors were invited to resubmit after an appeal against the decision.]

Thank you for submitting your work entitled "NG2 glial cells integrate synaptic input in global and dendritic calcium signals" for consideration by *eLife*. Your article has been reviewed by two peer reviewers, and the evaluation has been overseen by Gary Westbrook as the Senior Editor and Reviewing Editor. The two reviewers have opted to remain anonymous.

Our decision has been reached after consultation between the reviewers. Based on these discussions and the individual reviews below, we regret to inform you that your work will not be considered further for publication in *eLife*.

The manuscript addresses and important and topical issue. However, the impact of the work is somewhat reduced by a previous study that showed that VGCCs regulate NG2 calcium signals (Haberlandt et al., 2011). The authors' novel "mock PSP" approach is of interest, but the conclusions are limited to the role of A-type potassium channels modulation of VGCCs and the visualization of calcium signaling in NG2 cell processes. A more thorough characterization of mechanisms underlying NG2 cell calcium signaling and a more physiological stimulation paradigm (e.g. stimulation of synaptic input to NG2 cells) would increase the significance. Overall we think that these concerns will require substantial additional experiments that would exceed the *eLife* benchmark of 2 months. The full reviews are included below.

Reviewer #1:

The study by Sun et al. investigates a very important question in the field of neuron-glia communication. It is now well established that NG2 cells receive glutamatergic or GABAergic synaptic inputs in different regions of the brain, including both grey and white matter. However, how these synaptic inputs are encoded in these cells, and how synaptic responses are integrated intracellularly is still completely unknown. Furthermore, NG2 cells display significant cell process extension and most of the synapses are located on these processes. Therefore, it has been postulated that NG2 cell processes are major sites of synaptic integration in NG2 cells, but this has never been demonstrated.

Across glial types, intracellular Ca^2^ increase has been shown to be an established response signal to external input – from other glial cells or from neighboring neurons. In this vein, it would be logical to propose that NG2 cells also display robust Ca^2^ signals in response to synaptic activity. However, Ca^2^ signals in NG2 cells in the context of synaptic communication have not previously been systematically characterized. Sun et al. have attempted to do precisely this in their report.

Overall, the study is a detailed characterization of NG2 cell responses to synaptic inputs performed by using advanced experimental approaches. The biophysical results should be complemented by further analysis to broaden the physiological and neurobiological impact of this excellent study.

The entire study is performed by using either 1. Photo-release/2-photon uncaging of glutamate on different subdomains of NG2 cells or 2. Current injection in NG2 cells to elicit "mock PSPs". While these techniques are valid and important, they provide indirect evidence of a synaptic event. It would be very important to determine whether the observed Ca^2^ responses and the physiological role of A-type K channels are reproduced upon activation of glutamatergic synapses on NG2 cells by neuronal stimulation, i.e. under a more physiological stimulation paradigm. Does use-dependent inactivation of A-type channels occur under physiological (axonal) activation of NG2 cells?

A further important question is whether A-type K channels also play a modulatory role in NG2 cells of the white matter, i.e. whether this is an intrinsically conserved function of all NG2 cells in different brain regions.

Major points

1) Discussion. The Discussion is too long and does not include the crucial point of the potential physiological and developmental role of the physiological mechanisms investigated in this study. Significant advance has been made in our understanding of the potential physiological/developmental role of glutamatergic and GABAergic synapses on NG2 cells. Therefore, the authors should discuss more extensively the role of A-type channels in the context of: i) Glu-mediated synaptic transmission on NG2 cells, and ii) the role of A-type K channels in cell cycle regulation/cell proliferation, as Glu synapses have been shown to regulate NG2 cell proliferation (Mangin et al., 2012).

2) Abstract. The Abstract should be more specific, in particular should include the information that A-type K channels control regulation of intracellular Ca^2^ levels upon synaptic activity in NG2 cells.

3) Ca^2^ Imaging: It is laudable that the authors have recorded Ca^2^ increases at thin NG cell processes. It is also understandable that SNR (signal to noise ratio) for Ca^2^ imaging experiments conducted at thin cell processes such as distal dendrites may be relatively low. However, in the context of synaptic communication, it is important that the authors address this. This is especially problematic when drawing conclusions from imaging traces beyond a binary result (responsive vs. non-responsive cells or dendrites). Example: In Figure 3 – there does appear to be a clear increase in fluorescence in 3E "d" upon current injection. However, the basal calcium trace i.e. pre-current injection also seems to have changes in ΔF/F. This can also be seen in Figure 4 – "2.5 um" vs 32.5 "um" [What ROI does the "32.5 um" trace correspond to in the panel?]. Does this unstable ΔF/F signal constitute "basal" Ca^2^ activity? Alternatively, is this simply noise or a result of averaging lesser number of dendritic pixels? Is the definition of a "spikelet" – also based on an objective analysis or is it based on visual inspection of the trace?

4) Ca^2^ kinetics: Since the authors have used the template based method to detect calcium responses (Clements and Bekkers, 1997), it is appropriate to report and perform further analysis on the temporal aspects of the Ca^2^ response.

i) Although the authors mention that the response decays "on the order of many seconds", they never mention if they have performed a detailed analysis of the decay constant, which they mention in passing (under sub-section "Estimating peak free calcium concentration of mock PSP induced signals in NG2 cells"). How was the decay quantified? Given that in many representative traces, the ΔF/F signal does not return to baseline (Figure 3 "d", 4 D more obvious in "12.5 um" and "32.5 um", is there substantial variability in the decay constant? Similar to the claim that Ca^2^ onset time is more or less constant across distance, does the decay constant also remain the same across distance from the NG2 cell body? Since the authors distinguish NG2 cell dendritic Ca^2^ response from neuronal Ca^2^ kinetics (– Discussion section), it would be appropriate to show a neuronal dendrite Ca^2^ response as an inset in Figure 4.

ii) In all the Ca^2^ traces displayed, only single responses are shown. Have the authors attempted to elicit consecutive Ca^2^ events at dendrites?

Reviewer #2:

Calcium signaling has been shown to regulate many aspects of NG2 cell behavior from proliferation to differentiation and myelination. However, the link between synaptic input to NG2 cells and modulation of their behavior remains unclear. This manuscript seeks to add an important piece to the puzzle, directly linking depolarization via synaptic inputs to increases in intracellular calcium signaling in NG2 cells. The authors used clever approaches to simulate the effects of synaptic inputs to NG2 cells (mock PSPs) by injecting current. The authors discovered that A-type potassium channels modify intracellular calcium events mediated via voltage-gated calcium channels (VGCCs). They also explored calcium dynamics in the processes (and somata) of NG2 cells for the first time. These studies add to our understanding of how VGCCs regulate calcium events in both the somata and processes in NG2 cells and how A-type potassium channels modulate their activity.

Major Points

1) Intracellular calcium rises can be initiated by several potential mechanisms: i) Influx of extracellular calcium, ii) voltage or iii) ligand gated channels, and iv) release from internal stores.

i) the role of the Na -Ca^2^ exchanger (NCX) in NG2 cell calcium signaling remains controversial (Ge et al., 2006; Tong et al., 2009; Haberlandt et al., 2011). The authors should at minimum discuss the potential role of NCX.

ii) This manuscript focuses on the contribution of voltage-gated A-type potassium channels via inhibition by extracellular applied 4-AP. Since TEA does not affect mock PSP amplitude or duration, it would be helpful to confirm the effects on these parameters are mediated by A-type potassium channels are blocked by internal Cs . This would rule out effects on neuronal A-type potassium channels (and subsequently properties of spontaneous release onto NG2 cells) and also potential direct effects of 4-AP on VGCCs (Wu et al., 2009, PMCID: PMC2794761).

iii) The authors state that the contribution of ligand-gated channels to NG2 cell calcium elevations was minimal as 4-AP boosted the responses and their uncaging protocol did not elicit calcium signals (unpublished observations). A branch specific uncaging protocol may not activate enough glutamate receptors to cause a calcium event; the authors should use full field photo-uncaging to test whether glutamate receptor activation during control conditions can cause NG2 an intercellular calcium increase.

iv) The extended duration of NG2 calcium signals (2.5 seconds) suggests involvement of internal calcium stores. Indeed, Haberlandt et al., 2011 showed that thapsigargin reduced depolarization induced NG2 calcium signaling. The authors should at minimum comment on this in the Discussion section.

2) Whether stimulation of synaptic input to NG2 cells results in calcium events remains controversial. While Velez-Fort et al., 2010 report did not elicit somatic increases in calcium in the barrel cortex, Haberlandt et al., 2011 did document tetanic stimulation resulted in somatic calcium increases in some NG2 cells in the hippocampus (Figure 8, though responses were variable). The authors should attempt similar experiments to examine whether this type of synaptic stimulation results in increases in calcium in NG2 cell processes. Additional experiments using full field photo-uncaging or application of hypertonic aCSF would also greater contribute to determining whether stimulation of synaptic inputs results in NG2 cell calcium signaling under control conditions.

3) Previous studies have explored the involvement of calcium channel subtypes in mediating the calcium events in NG2 cells (Fulton et al., 2010; Haberlandt et al., 2011). Pharmacological experiments using the author's novel protocol would further strengthen the manuscript.

---

## [Author Response]

[Editors’ note: the author responses to the first round of peer review follow.]

Reviewer #1:

*[...] The entire study is performed by using either 1. Photo-release/2-photon uncaging of glutamate on different subdomains of NG2 cells or 2. Current injection in NG2 cells to elicit "mock PSPs". While these techniques are valid and important, they provide indirect evidence of a synaptic event. It would be very important to determine whether the observed Ca^2^ responses and the physiological role of A-type K channels are reproduced upon activation of glutamatergic synapses on NG2 cells by neuronal stimulation, i.e. under a more physiological stimulation paradigm. Does use-dependent inactivation of A-type channels occur under physiological (axonal) activation of NG2 cells?*

**Response:** We agree with the reviewer that the manuscript would be strengthened by using a more physiological stimulation paradigm. Therefore, we have now included data showing calcium signals in NG2 cells upon synaptic stimulation. Local synaptic stimulation produces calcium signals in dendrites which are comparable to the uncaging- and current injection-induced calcium signals reported so far.

These experiments were performed during pharmacological inhibition of calcium stores and thus provide further evidence that calcium stores are not essential for the calcium signal reported here which is caused by voltage-gated calcium channels.

We are not yet showing use-dependent inactivation of A-type channels by synaptic input as these experiments, as detailed below, are technically challenging and would have exceeded the time available for the revision. Our trial experiments suggest that the stimulation of two independent but nearby synaptic inputs will be required. One input will inactivate A-type channels but its vesicle pool will be exhausted thereafter. The second input can then serve as test input to elicit a calcium response without previous activation of input 1. Because both inputs need to target the same dendritic segment this is a difficult and time consuming experiment which we feel would possibly be most appropriate for a follow-up manuscript.

Nevertheless, the experiment shown in Figure 10 does show that at least mock PSPs inactivate A-type channels, which partly addresses the question of the reviewer (we have shown that mock PSPs are comparable in size and kinetics to real synaptic input).

*A further important question is whether A-type K channels also play a modulatory role in NG2 cells of the white matter, i.e. whether this is an intrinsically conserved function of all NG2 cells in different brain regions.*

**Response:** So far the main characteristics of NG2 cells have been found to be similar across many brain regions including white and grey matter. However, there are slight systematic differences between white and grey matter, with NG2 cells in the white matter showing a higher input resistance (Chittajallu et al., 2004) making the recruitment of voltage gated channels even more likely. We believe a follow-up study would be best suited for systematically exploring differences between white and grey matter.

*Major points*

*1) Discussion. The Discussion is too long and does not include the crucial point of the potential physiological and developmental role of the physiological mechanisms investigated in this study. Significant advance has been made in our understanding of the potential physiological/developmental role of glutamatergic and GABAergic synapses on NG2 cells. Therefore, the authors should discuss more extensively the role of A-type channels in the context of:*

i) Glu-mediated synaptic transmission on NG2 cells,

**Response:** We have dramatically shortened and sharpened our discussion to better shine light on the role of A-type channels in NG2 cells for their developmental response and behavior. In particular the following new paragraph addresses this point:

“Ca^2^ signals in NG2 cells are restrictively gated by A-type potassium channels. […] This adaptation likely is required to match the responsiveness of NG2 cells to patterns of activity which vary across regional neuronal circuits throughout the brain (Sun and Dietrich, 2013).”

*ii) the role of A-type K channels in cell cycle regulation/cell proliferation, as Glu synapses have been shown to regulate NG2 cell proliferation (Mangin et al., 2012).*

**Response:** The above paragraph addresses the potential role of A-type channels in regulating cell cycle or proliferation. However, to our knowledge there is so far no published data addressing the possible role of A-type potassium channels in regulating the cell cycle or proliferation of NG2 cells available. The series of papers by Gallo and coworkers did not look at the role of A-type channels as they found that cultured OPCs did not survive 4-AP treatment (Gallo et al., Ghiani et al., Knutson et al., Pende et al).

*2) Abstract. The Abstract should be more specific, in particular should include the information that A-type K channels control regulation of intracellular Ca^2^ levels upon synaptic activity in NG2 cells.*

**Response:** We have revised our Abstract as suggested.

“[…]Synaptic activity induces rapid Ca^2^ signals mediated by low-voltage activated Ca^2^ channels and under strict inhibitory control of voltage-gated A-type K channels[…] “and “[…]Taken together, the activity-dependent control of Ca^2^ signals by A-type channels and the global versus local signaling domains make intracellular Ca^2^ in NG2 cells a prime signaling molecule to transform neurotransmitter release into activity-dependent myelination.”

*3) Ca^2^ Imaging: It is laudable that the authors have recorded Ca^2^ increases at thin NG cell processes. It is also understandable that SNR (signal to noise ratio) for Ca^2^ imaging experiments conducted at thin cell processes such as distal dendrites may be relatively low. However, in the context of synaptic communication, it is important that the authors address this. This is especially problematic when drawing conclusions from imaging traces beyond a binary result (responsive vs. non-responsive cells or dendrites). Example: In Figure 3 – there does appear to be a clear increase in fluorescence in 3E "d" upon current injection. However, the basal calcium trace i.e. pre-current injection also seems to have changes in ΔF/F. This can also be seen in Figure 4 – "2.5 um" vs 32.5 "um" [What ROI does the "32.5 um" trace correspond to in the panel?]. Does this unstable ΔF/F signal constitute "basal" Ca^2^ activity? Alternatively, is this simply noise or a result of averaging lesser number of dendritic pixels?*

**Response:** The small upward deflection seen pre-current injection in 3E-d referred to by the reviewer represents classical noise as frequently appearing in PMT-based detection of weak light signals, when the overall photon count is low. This can clearly be seen in Figure 5 (formerly 4D) – "2.5 um" vs 32.5 „um“: these spurious spike-like “events” occur when normalizing the trace on a low pre-stimulus baseline which has been recorded from a small distal process with weak emission by Fluo-4. We now have included identical recordings from neurons (Figure 5) which show qualitatively the same noise. However, as the signals in neurons are larger the noise is less obvious. How the visibility of the noise increases with decreasing photon count can also nicely be seen along the neuronal dendrite (Figure 5, 2.5 µm -> 32.5 µm). We are confident that these spurious ∆F/F changes do not reflect basal calcium activity.

We have simplified the ROI labelling in the new version of the figure. The 32.5 µm trace corresponds to ROI 7 (7 * 5 µm increments minus 2.5 µm half of increment/ROI length).

*Is the definition of a "spikelet" also based on an objective analysis or is it based on visual inspection of the trace?*

**Response:** The term “spikelet” refers to the electrical recording in which we have seen this type of response when A-type channels had been blocked in an all-or-nothing fashion, so strictly speaking it was based on visual inspection.

We now have included a quantitative analysis of the spikelets which we observed in 5/7 NG2 cells upon injection of the gaussian-shaped current in the presence of 4-AP and report the amplitude to be 40.2 ± 6.7 mV.

4) Ca^2^ kinetics: Since the authors have used the template based method to detect calcium responses (Clements and Bekkers, 1997), it is appropriate to report and perform further analysis on the temporal aspects of the Ca^2^ response.i) Although the authors mention that the response decays "on the order of many seconds", they never mention if they have performed a detailed analysis of the decay constant, which they mention in passing (under sub-section "Estimating peak free calcium concentration of mock PSP induced signals in NG2 cells"). How was the decay quantified? Given that in many representative traces, the ΔF/F signal does not return to baseline (Figure 3 "d", 4 D more obvious in "12.5 um" and "32.5 um", is there substantial variability in the decay constant? Similar to the claim that Ca^2^ onset time is more or less constant across distance, does the decay constant also remain the same across distance from the NG2 cell body?

**Response:** In the new Figure 4, panel C now provides longer recordings of calcium signals in which the decay to baseline is clearly seen. We have quantified the decay time constant from these new recordings by fitting an exponential function (𝜏_decay_ = 3.2 ± 0.5 s, n = 9).

We would not like to draw firm conclusions on the decay time across distance from soma. The signal size and the recording duration of the frame scan experiments (Figure 5) suggest the decay is at least on the same order of magnitude but do not allow quantitative assessment.

The calcium signals obtained in line scan mode were specifically acquired from proximal dendrites (~7-10 µm) and do not yield the required spatial information.

*Since the authors distinguish NG2 cell dendritic Ca^2^ response from neuronal Ca^2^ kinetics (Discussion section), it would be appropriate to show a neuronal dendrite Ca^2^ response as an inset in Figure 4.*

**Response:** Following the reviewer’s suggestion we have recorded CA1 neurons and performed an identical experiment to compare signal kinetics, SNR and how neuronal calcium signals back propagate into dendrites, see Figure 5.

*ii) In all the Ca^2^ traces displayed, only single responses are shown. Have the authors attempted to elicit consecutive Ca^2^ events at dendrites?*

**Response:** We have now included new data where we demonstrate the consecutive reproducibility of the calcium signal in NG2 cells over 12 min of recording time (Figure 4).

*Reviewer #2:*

1) Intracellular calcium rises can be initiated by several potential mechanisms: i) Influx of extracellular calcium, ii) voltage or iii) ligand gated channels, and iv) release from internal stores.

*i) the role of the Na -Ca^2^ exchanger (NCX) in NG2 cell calcium signaling remains controversial (Ge et al., 2006; Tong et al., 2009; Haberlandt et al., 2011). The authors should at minimum discuss the potential role of NCX.*

**Response:** We have conducted additional experiments and now demonstrate that NCX is not involved in the calcium signals we report here (Figure 4). This is in line with extra pharmacological experiments showing our calcium signals are mediated by T-/R-type calcium channels (Figure 4) and with the previous observations of the rapid rise times, the block by Cd/Ni and the absence of activation of ligand-gated receptors.

Furthermore, we now also show that calcium signals elicited by synaptic stimulation and by current injection are not dependent on calcium stores (Figure 10, thapsigargin) in agreement with our former conclusions.

Since Haberlandt et al. reported store-dependent signals upon depolarizing NG2 cells we have devoted an extra paragraph of the discussion on this apparent discrepancy:

“Haberlandt et al. reported that depolarization-induced Ca^2^ signals in NG2 cells are very sensitive to thapsigargin seemingly contradicting our findings (Haberlandt et al. (2011). […] Thus, it is conceivable that NG2 cells generally can generate store-dependent Ca^2^ signals but that store-dependent signaling is not readily recruited by synaptic activity studied here.”

*ii) This manuscript focuses on the contribution of voltage-gated A-type potassium channels via inhibition by extracellular applied 4-AP. Since TEA does not affect mock PSP amplitude or duration, it would be helpful to confirm the effects on these parameters are mediated by A-type potassium channels are blocked by internal Cs . This would rule out effects on neuronal A-type potassium channels (and subsequently properties of spontaneous release onto NG2 cells) and also potential direct effects of 4-AP on VGCCs (Wu et al., 2009, PMCID: PMC2794761).*

**Response:** We have now performed calcium imaging experiments where Cs was included in the patch pipette to block potassium channels from the intracellular side. Current injection strength was adjusted to compensate for the resulting higher input resistance of NG2 cells when perfused with Cs. These experiments show that also Cs potentiates the calcium signals in NG2 cells and thereby exclude that an effect of 4-AP on neuronal potassium channels (Cs did not reach neurons, applied from inside the NG2 cell) or on calcium channels in NG2 cells (Cs has never been shown to facilitate calcium channels) is required.

*iii) The authors state that the contribution of ligand-gated channels to NG2 cell calcium elevations was minimal as 4-AP boosted the responses and their uncaging protocol did not elicit calcium signals (unpublished observations). A branch specific uncaging protocol may not activate enough glutamate receptors to cause a calcium event; the authors should use full field photo-uncaging to test whether glutamate receptor activation during control conditions can cause NG2 an intercellular calcium increase.*

**Response:** We seem to have caused a misunderstanding. We did not want to imply that a calcium increase in NG2 cells via glutamate receptors is not possible under any circumstances. Rather we wanted to point out that with a weak, branch-specific glutamate uncaging protocol we did not observe calcium increases via glutamate receptors alone – just as the reviewer would also not have expected it.

We now clarify this in the Discussion section:

“The kinetics and magnitude of Ca^2^ signals in NG2 cell dendrites induced by local synaptic stimulation or local glutamate uncaging match those evoked by somatic current injection very well suggesting that they are also caused by recruitment of VGCCs. […] However, this does not imply that glutamate receptors cannot be a source of Ca^2^ entry if more widespread opening of receptors is achieved (see Ge et al. 2006).”

*iv) The extended duration of NG2 calcium signals (2.5 seconds) suggests involvement of internal calcium stores. Indeed, Haberlandt et al., 2011 showed that thapsigargin reduced depolarization induced NG2 calcium signaling. The authors should at minimum comment on this in the Discussion.*

**Response:** We now experimentally show that calcium stores are not involved in the calcium signals we report: Both synaptically- and mock PSP-induced calcium signals are seen in the presence of thapsigargin (Figure 10). We also discuss potential differences to the Haberlandt study:

"Haberlandt et al. reported that depolarization-induced Ca^2^ signals in NG2 cells are very sensitive to thapsigargin seemingly contradicting our findings (Haberlandt et al. (2011). However, the aforementioned study used a much stronger stimulus and depolarized NG2 cells for 100 ms to +20 mV (voltage clamp) and the resulting Ca^2^ signal showed kinetics an order of magnitude slower than reported here. Thus, it is conceivable that NG2 cells generally can generate store-dependent Ca^2^ signals but that store-dependent signaling is not readily recruited by synaptic activity studied here."

*2) Whether stimulation of synaptic input to NG2 cells results in calcium events remains controversial. While Velez-Fort et al., 2010 report did not elicit somatic increases in calcium in the barrel cortex, Haberlandt et al., 2011 did document tetanic stimulation resulted in somatic calcium increases in some NG2 cells in the hippocampus (Figure 8, though responses were variable). The authors should attempt similar experiments to examine whether this type of synaptic stimulation results in increases in calcium in NG2 cell processes. Additional experiments using full field photo-uncaging or application of hypertonic aCSF would also greater contribute to determining whether stimulation of synaptic inputs results in NG2 cell calcium signaling under control conditions.*

**Response:** In Figure 10 we now show that synaptic stimulation elicits the same type of calcium signal as mock PSPs do (kinetics and amplitude). We have used local stimulation close to a dendrite and obtained local dendritic calcium responses using a substantially briefer stimulus when compared to the Haberlandt et al. study (6 vs 100 stimuli).

*3) Previous studies have explored the involvement of calcium channel subtypes in mediating the calcium events in NG2 cells (Fulton et al., 2010; Haberlandt et al., 2011). Pharmacological experiments using the author's novel protocol would further strengthen the manuscript.*

**Response:** We have included a pharmacological characterization and can now show that the calcium signals in NG2 cells are largely mediated by T- and R-type calcium channels using the blockers SNX-482 and TTA-P2.